# An integrated organoid omics map extends modeling potential of kidney disease

Moritz Lassé [1,2], Jamal El Saghir [3], Celine C. Berthier [3], Sean Eddy [3], Matthew Fischer [3], Sandra D. Laufer [1,2], Dominik Kylies[1,2], Arvid Hutzfeldt [1,2], Léna Lydie Bonin [4], Bernhard Dumoulin[1,2], Rajasree Menon[5], Virginia Vega-Warner [3], Felix Eichinger[3], Fadhl Alakwaa [3], Damian Fermin [3], Anja M. Billing [4], Akihiro Minakawa [3], Phillip J. McCown [3], Michael P. Rose [3], Bradley Godfrey[3], Elisabeth Meister [1,2], Thorsten Wiech [2,6], Mercedes Noriega [2,6], Maria Chrysopoulou [4], Paul Brandts [1,2], Wenjun Ju[3], Linda Reinhard [1,2], Elion Hoxha[1,2], Florian Grahammer[1,2], Maja T. Lindenmeyer [1,2], Tobias B. Huber [1,2], Hartmut Schlüter [7], Steffen Thiel [4], Laura H. Mariani[3], Victor G. Puelles [1,2,8,9], Fabian Braun [1,2], Matthias Kretzler [3,5], Fatih Demir [4], Jennifer L. Harder [3,11] ✉ & Markus M. Rinschen [1,2,4,10,11] ✉

Kidney organoids are a promising model to study kidney disease, but their use is constrained by limited knowledge of their functional protein expression profile. Here, we define the organoid proteome and transcriptome trajectories over culture duration and upon exposure to TNFα, a cytokine stressor. Older organoids increase deposition of extracellular matrix but decrease expression of glomerular proteins. Single cell transcriptome integration reveals that most proteome changes localize to podocytes, tubular and stromal cells. TNFα treatment of organoids results in 322 differentially expressed proteins, including cytokines and complement components. Transcript expression of these 322 proteins is significantly higher in individuals with poorer clinical outcomes in proteinuric kidney disease. Key TNFα-associated protein (C3 and VCAM1) expression is increased in both human tubular and organoid kidney cell populations, highlighting the potential for organoids to advance biomarker development. By integrating kidney organoid omic layers, incorporating a disease-relevant cytokine stressor and comparing with human data, we provide crucial evidence for the functional relevance of the kidney organoid model to human kidney disease.

Organoids are emerging as an increasingly important model system to understand development and disease. Kidney organoids have been used to model kidney cancer, glomerular diseases and podocytopathies, basement membrane development and disease, polycystic kidney disease (PKD), and renal tubular epithelia ciliopathies[1–7]. Additionally, organoids have been proposed as a promising screening tool for therapeutics, as well as a model of virus infection and organ cryopreservation, especially when combined with high throughput methods and organ-on-a-chip microfluidics that allow mechanical forces to be applied[1,4,8–13]. Organoids are employed to improve our understanding of genetic kidney diseases and are used as models to interrogate kidney development and molecular mechanisms[14–17]. When

generated from human pluripotent stem cells, kidney organoids recapitulate elements of human kidney disease which are lacking in animal models and traditional two-dimensional (2D) tissue culture models[18].

Significant effort by us and others[1,19–23] has defined single-cell transcriptional profiles of kidney organoids. However, transcripts do not necessarily correspond to protein abundance, especially in very dynamic systems[24]. In addition, several dimensions of the proteome are not amenable to transcript-based analysis, including the secretome, which is the entirety of secreted proteins. Until now, the proteome of human kidney organoids has been insufficiently characterized; it is still unclear how their proteome changes during differentiation, and whether the protein composition of kidney organoids is sufficient to model more complex disease processes such as that seen, for instance, in inflammatory tissue responses. Through recent advances in large-scale proteome acquisition technologies, novel perspectives on tissue biology in kidney disease have been gained. We can harness these novel technologies and insights to both define the proteome of kidney organoids more fully, and to further evaluate their relevance to human kidney disease.

Using transcriptional profiling, we recently demonstrated that proinflammatory molecular signals, likely orchestrated through cytokines such as TNFα, were observed in kidney tissue from individuals with poor clinical outcomes in proteinuric kidney disease. For many kidney diseases, individualization of therapy is challenged by diverse underlying pathomechanisms[25]. Thus, a more exhaustive proteomics analysis of kidney organoids would further enhance our understanding of proinflammatory events associated with these putative inflammatory drivers of disease.

The data we present here provide critical foundational knowledge of kidney organoids as a model system of human kidney disease. First, we describe how the organoid proteome and transcriptome evolve as a function of culture duration and identify organoid cell types in which protein expression changes. Then we compare the expression of podocyte-specific proteins in organoids to the expression observed in native glomeruli and cultured podocyte cells. Finally, we demonstrate that organoids react to TNFα with a global inflammatory response similar to the molecular changes associated with poor outcomes for individuals with proteinuric kidney diseases focal segmental glomerulosclerosis (FSGS) or minimal change disease (MCD). These protein signatures can directly add to an individual's disease stratification and suggest additional disease-relevant biomarkers. Together, this characterization of kidney organoids based on integration of proteome and transcriptome along with the demonstration of innate immune responses in organoid cell types expands the scope of organoids as a discovery platform for the cellular biology of kidney disease and potential therapeutics. Moreover, we provide a rich resource to the research community to promote ongoing kidney disease modeling, biomarker discovery and therapeutic screening.

## Results

### Kidney organoid protein expression changes with culture duration

Building on work generated by us and others showing that kidney organoids around 3 weeks in culture display glomerular differentiation characteristic of developing human kidneys[19,23] we performed proteomic and transcriptional profiling[1,26] on organoids cultured between 21 and 29 days. Proteomic analysis was performed on 20 organoid spheroids in triplicate at four-time points during the culture period. In total, more than 6700 proteins were identified (Supplementary Data 1). Of these identified proteins, 5403 proteins could be quantified across samples. Principal component analysis revealed the separation of organoid proteomes from days 21 to 25 to 27 to 29 (Supplementary

Fig. 1A). We performed differential analysis of these 5403 proteins between days 29 and 21 of differentiation and detected 350 proteins that were significantly upregulated, and 428 proteins which were significantly downregulated (FDR < 0.05, Fig. 1a and Supplementary Data 2). Key podocyte markers indicating glomerular differentiation, such as nephrin (NPHS1) and synaptopodin (SYNPO), decreased with time in culture indicating a relative loss of podocytes or a relative increase of other cell populations (Fig. 1a). Meanwhile, structural proteins such as smooth muscle actin (ACTA2) and key regulatory differentiation proteins such as Platelet-derived growth factor receptor alpha (PDGFRA) were increased (Fig. 1a), suggesting increased production of extracellular matrix. These observations were validated by immunofluorescence (Fig. 1b). We also observed increased levels of collagen type I alpha 1 chain (COL1A1) and fibronectin type 1 (FN1) between days 21 and 29, which were also validated by immunofluorescence (Supplementary Fig. 1B, C). Single-cell transcriptional profiling suggested that these proteins were mostly generated by stromal cells (Fig. 2d, Supplementary Fig. 1D). To further visualize culture duration-dependent trajectories, we normalized protein expression data and performed row-wise hierarchical clustering (Fig. 1c) followed by GO-term annotation of the respective proteins and clusters. Five clusters (of size >25 proteins) were distinguished (Fig. 1d). The data confirmed the patterns of an increase in extracellular matrix proteins (Fig. 1d, clusters 1 and 3) and a decrease in glomerular development proteins (Fig. 1d, cluster 5) with culture duration.

### Organoid proteome-transcriptome integration uncovers cellular origins of proteins

To further assess gene expression dynamics in organoids relative to duration in culture, we integrated proteomic with bulk RNA sequencing data of corresponding organoid spheroids at days 21, 25, and 29 (Fig. 2a). Measured quantities of protein and RNA (Supplementary Data 3) did not correlate strongly ($R = 0.27$, 0.24, 0.24 on days 21, 25, 29, respectively), consistent with prior observations[24]. The relationship between protein and RNA expression remained consistent across the days in culture as indicated by 2D UniProt keyword enrichment of basic functional terms, including 'Differentiation', 'Glycolysis', 'Mitochondrion', 'TCA-cycle' and 'Protein-biosynthesis' (Fig. 2a, b). To understand which cell types contributed to the proteome, we sought to define the cellular composition of organoids by employing single cell RNA sequencing (scRNA-seq). Integrated clustering of organoid-specific single-cell datasets conducted within our study revealed 14 different organoid cell types, 60% of cells identified as target kidney cell types (Figs. 2c, 4a, Supplementary Data 4). Each of these cell types contributed to the data observed in bulk proteomic and bulk RNA sequencing analysis. To define which cell types may undergo the most dramatic changes in the cellular organization during cell culture on the protein level, we mapped transcript markers of these 14 cell clusters to the proteomics dataset. This approach is feasible as scRNA-seq and proteomics markers are largely consistent[27]. This analysis suggested that the largest proteome changes occur in the stromal cell population as indicated by the over-expression of proteins related to stromal markers on day 29 compared with day 21 (Fig. 2d). Of the kidney-specific cell types, early glomerular epithelial cells and maturing podocytes seem to become less represented with duration in culture as indicated by under-expression of associated markers at the later timepoint. This is consistent with our earlier findings showing diminished podocyte and increased stromal cell-specific protein expression (Figs. 1b–d, 2e).

To understand which proteins were most associated with these observed changes, we clustered protein and RNA expression data of the top 30 differentially expressed proteins by cell type marker. The pattern of expression of these 30 proteins corresponded to observed changes in both protein and RNA levels. The majority (≥90 %) of the top 30 differentially expressed proteins decreased over time on both

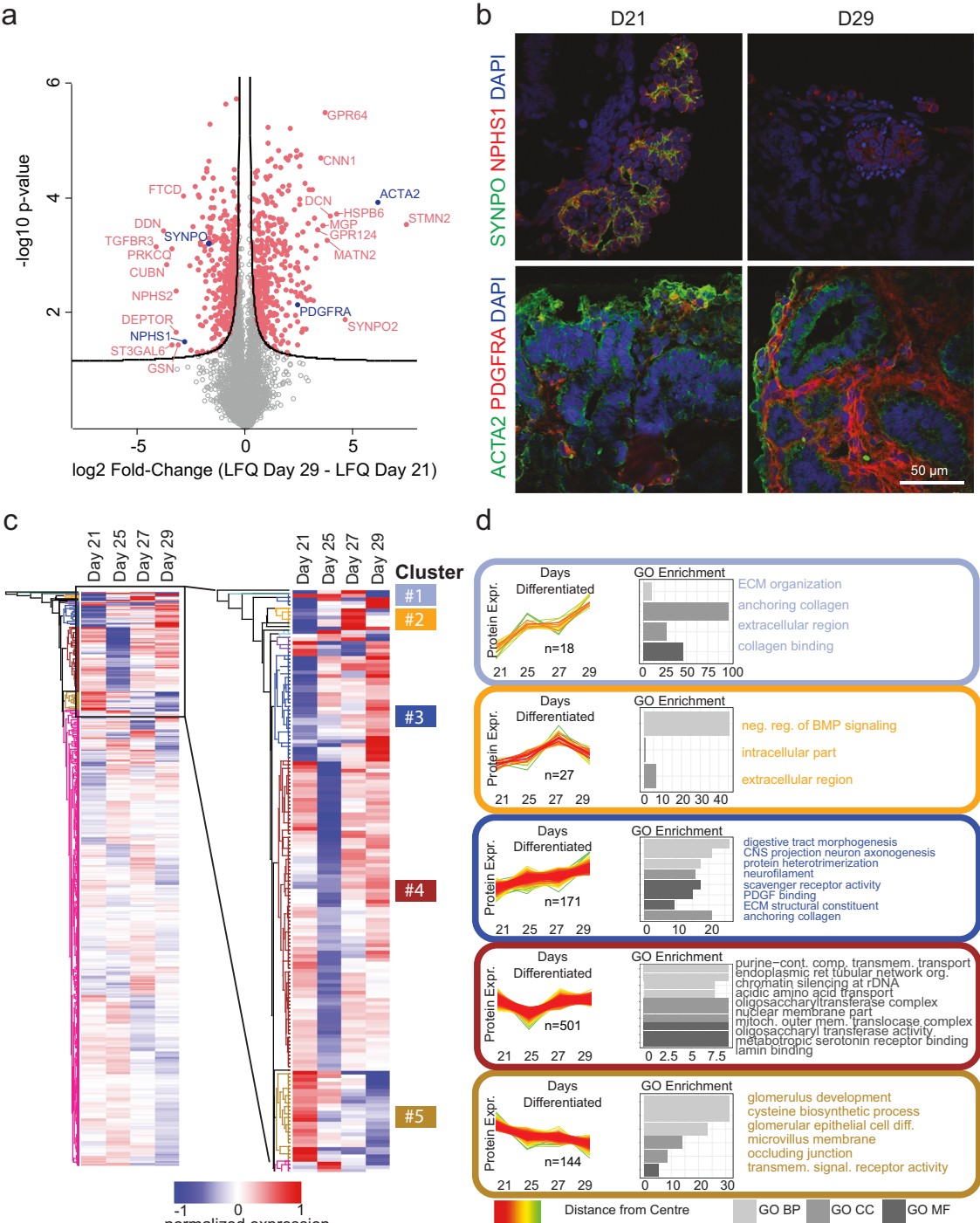

**Fig. 1 | Proteome of kidney organoids evolves with duration in culture.**
**a** Volcano plot of proteomic differential expression analysis (log₂ fold change of label-free quantification intensity comparing D29 with D21). Proteins are represented by dots and the black line illustrates the significance cut-off (two sided $t$ test, FDR < 0.05 and s0 = 0.1). Red colored proteins meet the significance threshold. Examples of strongly regulated proteins are labeled. Blue colored and labeled dots represent proteins highlighted in (**b**). **b** Immunofluorescence imaging of sectioned kidney organoids showing expression of (top panel) podocyte markers nephrin (*NPHS1*) and synaptopodin (*SYNPO*), and (bottom panel) cell structure and differentiation markers smooth muscle actin (*ACTA2*) and platelet-derived growth factor receptor alpha (*PDGFRA*), plus nuclear marker DAPI, $n = 3$, representative images shown, scale bar: 50 μm. **c** Heatmap (maximum distance) of normalized protein expression (mean subtracted label-free quantification values (average of three replicates)) during organoid differentiation. The zoomed-in region expands on the clusters that undergo marked changes during differentiation. Five clusters (of size >25 proteins) were distinguished and are indicated with colored rectangles. These five clusters are shown in (**d**). **d** GO enrichment analysis of unbiased clustering (clusters >25 proteins) of proteins during organoid differentiation (D21–D29) (Fisher's exact test, FDR < 0.05). Source data are provided as a Source Data file.

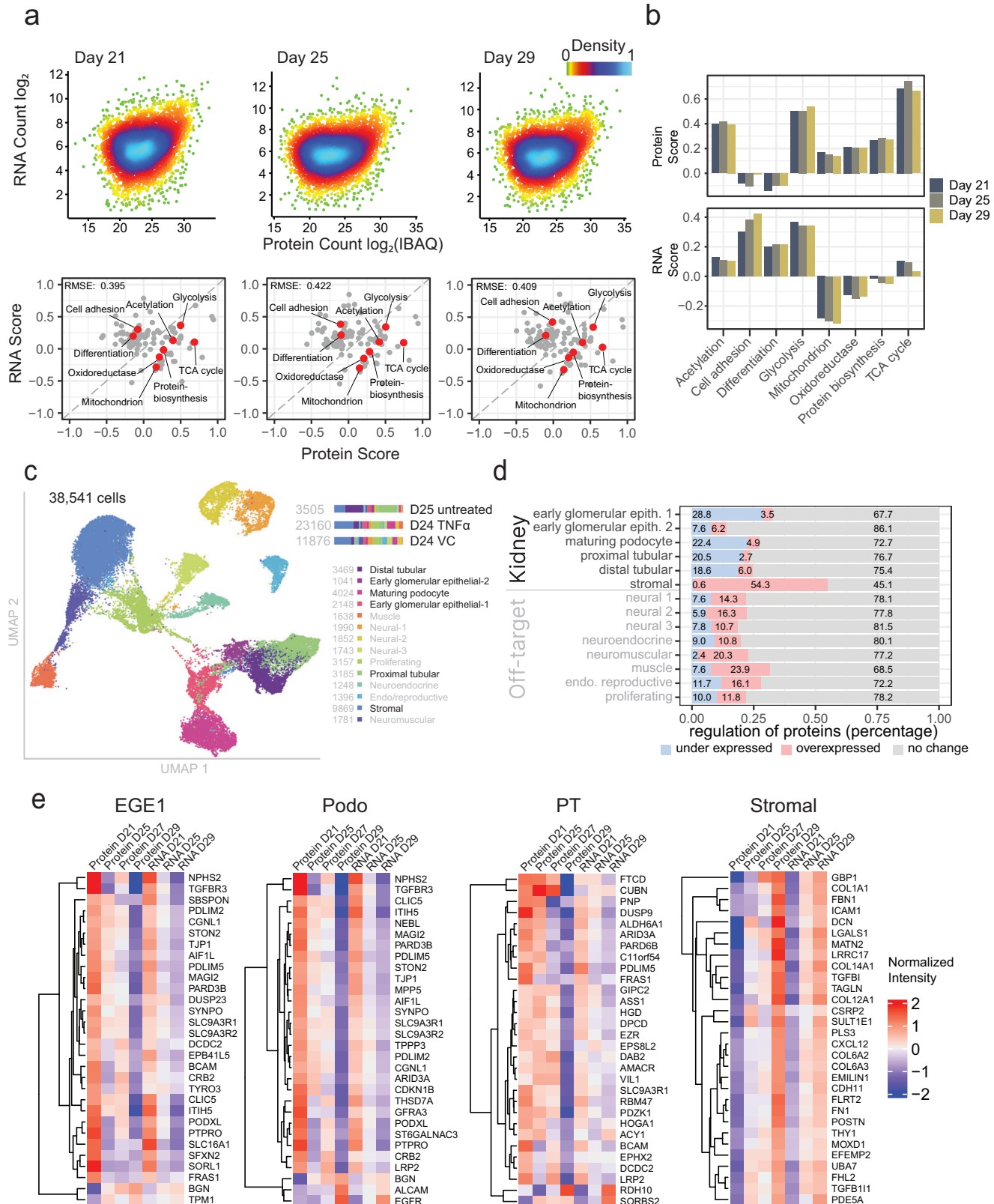

protein and RNA levels in the glomerular epithelial cells, podocytes and proximal tubular cells (Fig. 2e). This contrasted with stromal cells, where the top 30 differentially expressed proteins increased over time. Interestingly, several proteins demonstrated significant alteration in protein but not transcript expression level (Supplementary Fig. 2A). Pathway analysis of differentially expressed genes in day 29 relative to day 21 organoids predicted 107 upstream regulators (30 transcription

regulators), a list which prominently included highly enriched TGFB1 mechanistic networks (Supplementary Fig. 4F, Supplementary Data 3, 14, 16, 17). Together, these results suggest a loss of expression of kidney-specific proteins in older organoids accompanied by accumulation of fibrosis, highlighting the need to identify the optimal culture time point for expression of proteins of interest when modeling disease. Moreover, they reinforce the concept that transcript

**Fig. 2 | Integrated expression analysis of proteome-transcriptome trajectories over organoid culture duration. a** Scatterplot of RNA copy number (D21, D25, D29) and protein copy number with light blue color indicating high data point density and green for low density. Associated two-dimensional UniProt-keyword enrichment with basic functional terms, including 'Differentiation', 'Glycolysis', 'Mitochondrion', 'TCA-cycle' and 'Protein-biosynthesis' are highlighted in red. **b** Barplots of basic functional terms from 2D UniProt-keyword enrichment (from **a**). **c** Uniform Manifold Approximation and Projection (UMAP) representation of combined datasets from single cell transcriptomes of D25 untreated plus D24 TNFα-treated and vehicle control (VC) distinguished 14 cell type clusters. Visualization was carried out using Cell x Gene software; the contribution to the total cell number by each sample and cell type cluster is shown on the right. **d** The summed protein expression direction of corresponding transcript markers which were used to define the 14 cell clusters. Proteins were classified as overexpressed (FDR < 0.05 and log$_2$ fold-change >0), under-expressed (FDR < 0.05 and log$_2$ fold-change <0) or not differentially expressed (FDR ≥ 0.05) between D21 and D29. **e** Heatmap k-means clustering of bulk RNA and bulk protein of transcript cell-type markers for early glomerular epithelial 1 (EGE1), maturing podocyte (Podo), proximal tubular (PT) and stromal cells. The top 30 differentially expressed proteins D29 versus D21 (FDR < 0.01) were plotted. Source data are provided as a Source Data file.

expression does not necessarily inform quantitative alternations in protein expression.

## Comparison of organoid proteome organization with mature human kidney tissue

We next sought to determine how proteomes of organoids (over the culture period days 21–29), human kidney and immortalized podocytes overlapped. First, proteomes of organoids and microdissected adult single human tubules and single glomeruli were compared using a technology previously developed[28] (Fig. 3a). We achieved a six-fold deeper proteome coverage for organoids (n = 6703 proteins) compared to single glomeruli (n = 1002 proteins) or tubule native tissue extracts (n = 1730 proteins). Notably, the proteome from organoids covered 88% of the proteome of single glomeruli and 84% of the proteome of single tubules. GO-term over-representation analysis (Fig. 3b) indicated that the organoid proteome comprised of enriched gene sets corresponding to receptor-mediated signaling and 'Extracellular Matrix (ECM)' components. Some GO-terms were less represented in the organoids compared to human kidney components suggesting limitations to their modeling ability, including "glomerular vasculature" and "collagen type IV" as well as "brush border", "glucose:sodium- and urate-transport" representing proteins involved in classical proximal tubule function and solute transport.

Immortalized podocytes (Fig. 3c) demonstrated isolated proteins related to cell cycle and catabolic processes. Furthermore, the proteome copy numbers of key podocyte proteins in organoids were similarly distributed to those observed in mature native human and mouse podocytes, as compared to the relatively low numbers observed in immortalized podocytes (Fig. 3d). To visualize the three-dimensional environment and potential cellular crosstalk in the kidney organoids, we performed in silico interaction analysis using single-cell transcriptome data and NicheNet (Supplementary Fig. 2B). Specific interactions between expressed ligands on proteome level and cell-specific receptor programs were discovered. Together, these results indicate that kidney organoids faithfully represent the majority of proteins expressed in human kidneys, including key podocyte proteins, and are superior in this respect to immortalized podocytes.

## Organoids respond to TNFα by expressing proinflammatory proteins

In a recent study, we showed that TNFα-treated kidney organoids expressed key transcripts and proteins associated with TNFα activation and poor clinical outcomes in FSGS and MCD[25]. The expressed molecules included the chemokine ligand 2 (also known as monocyte chemoattractant protein 1, MCP-1, encoded by *CCL2*) and tissue inhibitor of metalloproteinases 1 (*TIMP1*). Building on insights of proteome organization of our organoid model (Figs. 1–3), we aimed to further define TNFα activation in kidney organoids. Though TNFα receptors TNFRSF1A and TNFRSF1B (encoded by *TNFRSF1A* and *TNFRSF1B*, respectively[29]) were not detected in organoids by proteomics, expression was detected in kidney cell types by scRNA-seq analysis at day 24, with *TNFRSF1A* expression far exceeding that of *TNFRSF1B*

(Fig. 4a, b, Supplementary Data 4). Further, immunofluorescence of day 25 organoids indicated strong expression of TNFRSF1A in podocytes (SYNPO+), proximal tubular cells (CDH2+) and stromal cells (MEIS1/2+) (Fig. 4c). Given these findings plus limited extracellular matrix deposition (Fig. 1b, Supplementary Fig. 1B, C), day 24-25 organoids were treated with TNFα to study their proteomic response (Fig. 4d, Supplementary Data 5).

After 24 h of TNFα stimulation, organoid cell lysates demonstrated a significant increase in Vascular cell adhesion protein 1 (VCAM1) concentration (FDR $p < 0.1$), which persisted to 48 h. After 48 h of TNFα treatment, 145 proteins were increased, and 157 proteins were decreased (FDR < 0.1) in the organoid proteome (Fig. 4d). Increased proteins included VCAM1, intercellular adhesion molecule 1 (ICAM1), Nuclear factor NF-kappa-B (NFKB2) and integrin alpha-3 (ITGA3). Protein markers of early glomerular epithelial cells were most altered with TNFα treatment compared with protein markers of other cell types (Fig. 4e). Stromal cells yielded the lowest response with TNFα treatment. We confirmed the localization of TNFα-induced ITGA3 and VCAM1 in podocytes (SYNPO+ or PODXL+), as well as colocalization of the TNFα receptor TNFRSF1A with VCAM1 (Supplementary Fig. 4A, B). While NFKB2 transcript expression was also significantly higher in podocytes following TNFα stimulation (Supplementary Fig. 4C), transcript expression of classic podocyte markers including WT1, NPHS1, and NPHS2 (Supplementary Fig. 4D) were significantly decreased suggesting podocyte stress in response to TNFα stimulation.

To further investigate possible signaling systems deployed by organoids upon TNFα exposure, we also analyzed the proteins secreted from organoid cells into the overlying culture medium, the so-called secretome. To define the medium at baseline, analysis without exposure to organoids was carried out and revealed a total of 23 proteins not commonly observed as contaminants as part of the MaxQuant contaminant database[30] (Supplementary Data 6). The organoids secreted an additional ~120 detectable proteins on average (Supplementary Fig. 3A). This number further increased following TNFα treatment (Supplementary Data 7); differentially expressed and secreted proteins included cell adhesion proteins but also regulators of apoptosis and cell death (SFRP2 and IGFBP7) and extracellular matrix proteins (LAMB1, HSPG2, PTX3 and BGN). Intriguingly, we found that TNFα treatment increased the secretion of the cytokine CXCL10 as well as several complement components (C1s, C3, C1q). Increased complement and cytokine expression was already visible after 24 h (Fig. 4f, Supplementary Data 7), while C1R and C1S expression was detected after 48 h of TNFα exposure. These secreted proteins were not found in the extracellular matrix (Geltrex) or in the acellular supernatants (Supplementary Fig. 3B). These analyses of the secretome demonstrate that kidney organoids (devoid of immune cell types) are capable of secreting proteins involved in cytokine signaling.

Importantly, TNFα-induced gene expression was robustly reproducible, as demonstrated by focused analysis of CXCL10 transcript as well as intracellular plus secreted protein levels following

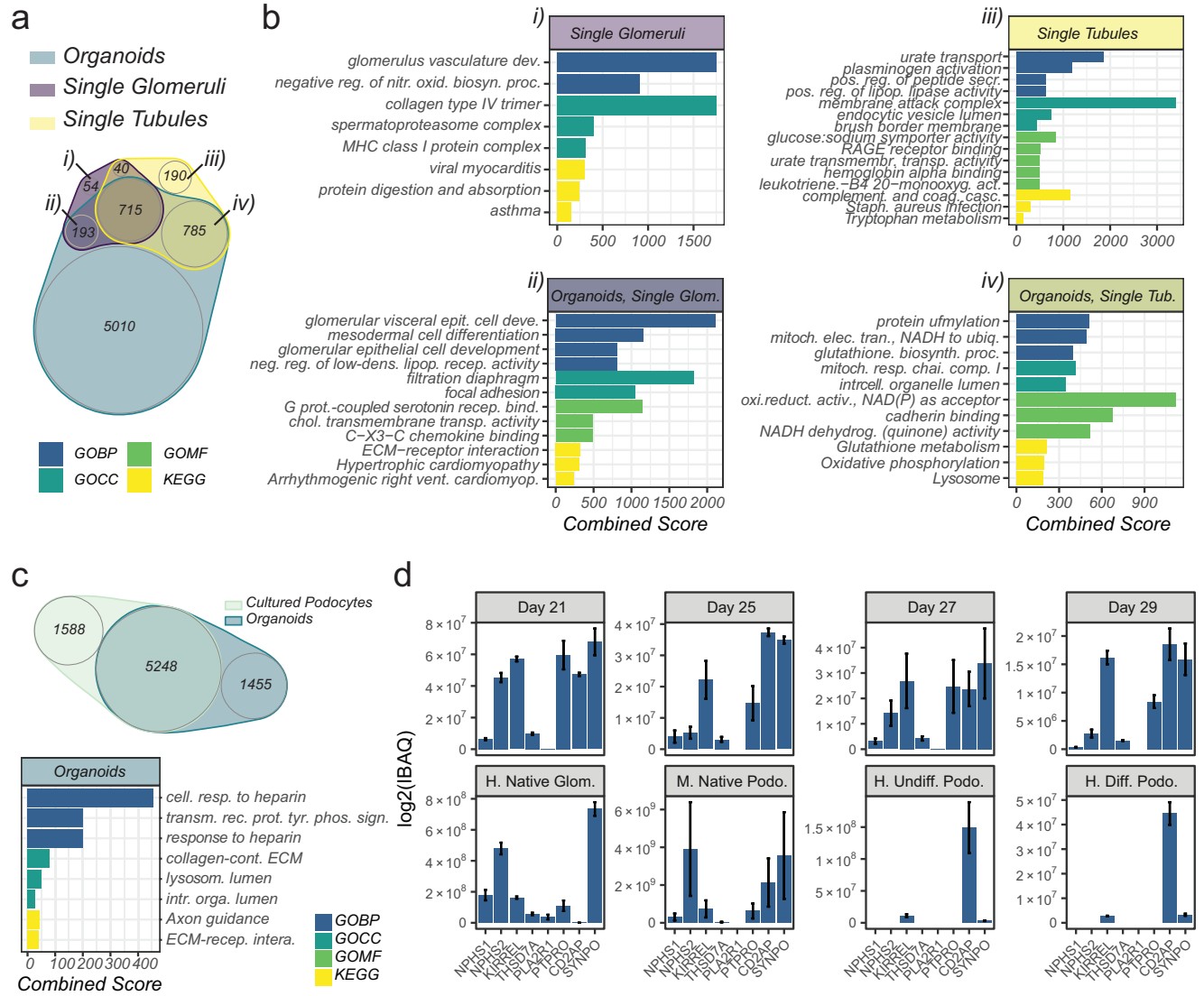

**Fig. 3 | Organoid proteome organization shows similarities with human kidneys. a** Venn diagram of the organoid proteome compared with microdissected single glomeruli and tubules proteomes. Four compartments indicate proteins identified in (i) single glomeruli only, (ii) both single glomeruli and organoids, (iii) single tubules only, and (iv) both single tubules and organoids. **b** Over-representation plots corresponding to Venn diagram compartments in (**a**) for Gene Ontology (GO) terms related to biological process (BP), molecular function (MF), cellular component (CC), and Kyoto Encyclopedia of Genes and Genomes (KEGG) pathways (Fisher's exact test, adjusted $p$-value < 0.05). The x-axis corresponds to

the combined score from EnrichR analysis. **c** Venn diagram (top) with associated over-representation plot (bottom) of terms uniquely mapping to the organoid proteome but not to cultured podocyte proteome. **d** Protein copy number based on intensity based absolute quantification (IBAQ) of podocyte markers expressed in organoids ($n = 3$) compared to human tissue ($n = 2$), mouse tissue ($n = 3$) and cultured human podocytes (undifferentiated and differentiated, both $n = 3$). Human (H); Mouse (M). Data are represented as mean ± SEM. Source data are provided as a Source Data file. Source data are provided as a Source Data file.

24 h and 48 h of TNFα exposure in multiple independent experiments (Fig. 4g, Supplementary Fig. 7B, D). Further, these proteins were not detectable in cultured podocytes treated with the same concentration of TNFα (Supplementary Data 8 and 9, Supplementary Fig. 3C–H), indicating that the protein expression profile of TNFα-treated organoids had a more complex biological response (particularly noticeable in the secretome) compared with the human cultured podocyte model.

Finally, we found that several of the induced proteins mapped on the canonical TNF pathway (hsa04668, Supplementary Fig. 4E), particularly after 48 h. Upstream regulator analysis suggested that there was significant enrichment for NF-kappa-B-induced genes in the TNFα treated samples with high consistency across sets of biological experiments (Supplementary Fig. 4G, Supplementary Data 15, 18).

## Proteome alterations in TNFα-induced organoids help stratify diseased human kidney tissue

We next sought to determine the translatability of the TNFα-driven changes observed in organoid proteomes to human kidney disease; expression of these genes was examined in tissue from individuals with MCD or FSGS within the NEPTUNE cohort who had poorer outcomes[25] (Fig. 5a). In that study, unsupervised clustering of kidney tissue transcriptomes from individuals with FSGS/MCD identified three sub-groups, one with poorer outcomes including loss of renal function (cluster 3). Computational analysis of genes differentially expressed in this subgroup's tissue relative to clusters 1 and 2 resulted in a 272-gene signature of TNF activity (Tissue TNF signature). We asked whether kidney organoids, containing kidney cells but no immune cells, could further focus this tissue-based signature to reveal kidney cell-specific pathomechanisms associated with poor outcome.

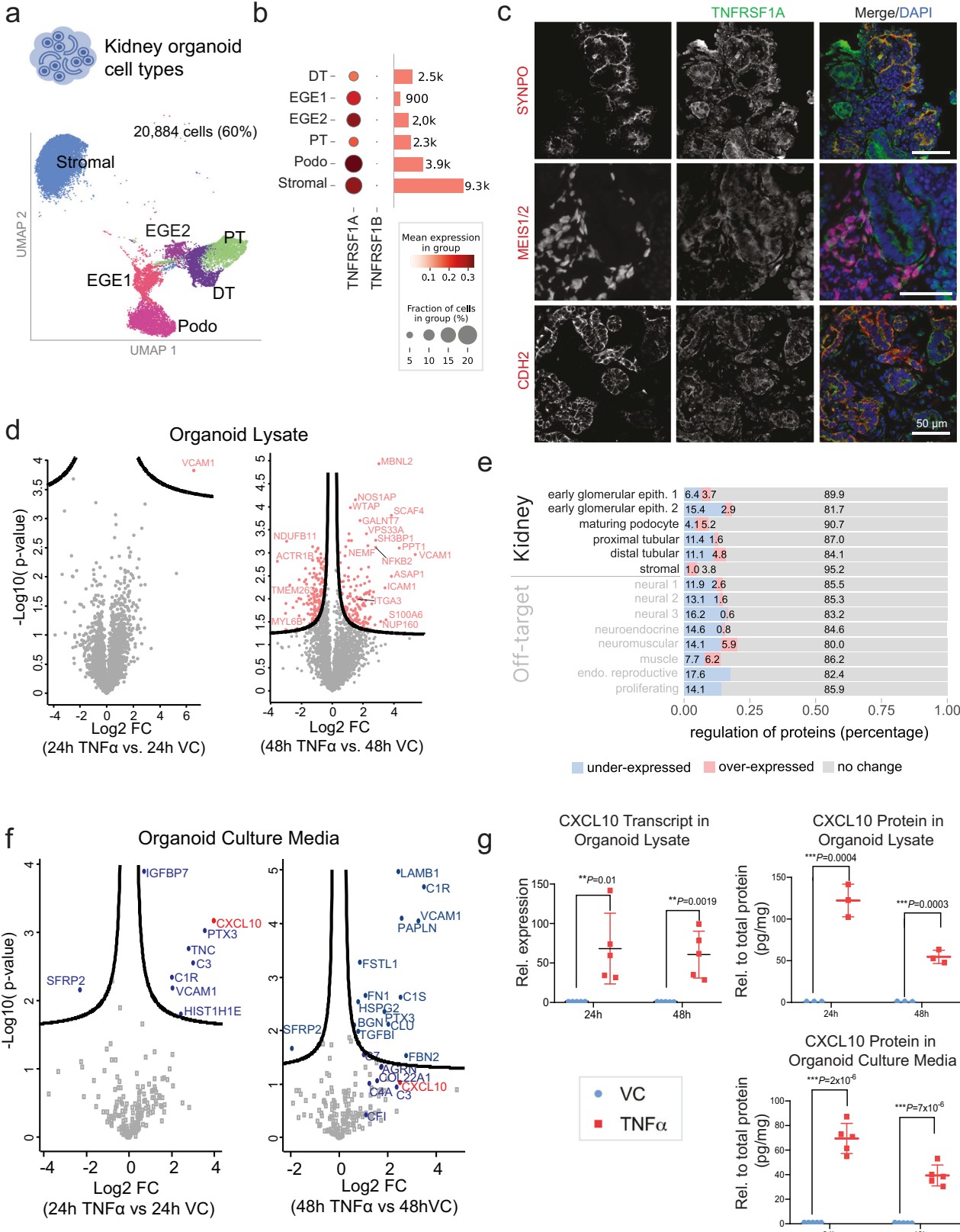

To explore this idea, a new 322-gene product signature (Organoid TNF signature) was derived from all differentially expressed proteins identified in TNFα-treated organoids (Fig. 4d, f, Supplementary Data 10, Fig. 5a). These TNF-dependent proteins had higher expression on gene level in various diseased kidney tissue compared to living donor tissue (Fig. 5b). Ten gene products were shared by the two gene

signatures as shown in Fig. 5c and Table 1. These included CXCL10 and cytokine response genes ICAM1, VCAM1, MAP4K4, PTX3 as well as complement factor C3. Most are known to be linked with TNFα-associated inflammation and a diverse range of kidney diseases as summarized in Table 1. Half of the ten proteins were identified in the secretome (gene names in red and blue in Fig. 5c), prioritizing

**Fig. 4 | TNFα treatment significantly alters protein expression and secretion in kidney organoids. a** UMAP highlighting 6 kidney cell type clusters identified in Fig. 2c, representing 60% of cells from a total of 35,036 analyzed organoid cells. **b** Dot plot showing transcript expression of TNFα-responsive receptors TNFRSF1A and TNFRSF1B in kidney cell clusters in (**a**). Bar length and associated numbers indicate the number of cells in the clusters. **c** Immunofluorescence imaging of sectioned kidney organoids showing expression of TNFRSF1A (*TNFRSF1A*) in co-staining with either synaptopodin (*SYNPO*, top row), stromal cell marker Meis Homeobox 1/2 (*MEIS1/2, middle row*), or tubular epithelial cell marker N-cadherin (*CDH2*, bottom row). DAPI, nuclear marker; *n* = 3, representative images shown, scale bar: 50 μm. **d** Differential proteome expression analysis (log₂ fold change of 24 h and 48 h) in cell lysates of day 25 TNFα-treated organoids compared with VC (two-sided *t* test, FDR < 0.1 and s0 = 0.1). **e** All quantified cell cluster marker proteins categorized into over-expressed, under-expressed or not differentially expressed upon TNFα stimulation 48 h vs VC. **f** Differential expression analysis (log₂ fold change) of 24 h and 48 h day 25 TNFα-stimulated organoid supernatant compared with VC. **g** Expression of *CXCL10* transcript and protein following treatment with recombinant TNFα for 24 h and 48 h. Data are represented as mean ± SEM of 5 independent experiments for qPCR and ELISA on culture media (top left and lower right graphs) and 3 independent experiments for ELISA on lysates (top right graph). Unpaired *t* test. VC, vehicle control. Source data are provided as a Source Data file.

potential biomarkers of TNFα activity. Importantly, pathway analysis of genes from both signatures independently and in combination (total 584 genes) revealed multiple instances of TNF network activity (Fig. 5d, Supplementary Data 11 and 12). However, the Organoid TNF signature revealed more about intracellular functions (amino acid and derivative metabolism, cellular response to stress, glycolysis, ATP synthesis), while the Tissue TNF signature encompassed more extra-cellular cytokine and immune system signaling. This suggests that the Organoid TNF signature is focused on gene activity at a more cell-based level as opposed to the complex milieu captured by the Tissue TNF signature. Each signature alone identified different aspects of gene connectivity based on literature-based network analysis (Supplementary Fig. 5A). This illustrates that complementary information can, indeed, be gleaned from adding the organoid model system to understand pathomechanisms of kidney disease. Performing analogous literature-based network analysis using all combined 584 genes and focused around the ten overlapping genes shows the unified interplay of these networks (Fig. 5d).

When the 322-gene Organoid TNF signature was applied to the same kidney-tissue transcriptome clusters as in Fig. 5a, significantly elevated summary transcript expression (Organoid TNF Score) was observed in kidney tissue of individuals from cluster 3 of the cohort for the 319 genes of the 322 gene score expressed (Fig. 5e, Supplementary Data 19). The signature score was underwhelming for the unsorted group of 220 individuals (Fig. 5e, far right bar). Similarly, only a modest performance of the signature score was observed when the Organoid TNF signature was applied to tissue transcriptomic data from other groups of kidney disease types (Supplementary Fig. 5B). This highlights the critical need to identify subgroups of individuals in disease cohorts who have different pathomechanisms of disease, and who may benefit from different targeted therapies.

When the 322-gene Organoid TNF signature was separated into signatures derived from either the secretome or the cell lysate proteome alone (two proteins were present in both), each signature could independently distinguish cluster 3 from clusters 1 and 2 on the transcriptional level in kidney tissue (Fig. 5f, Supplementary Data 19). Thus, both intracellular and secreted proteins expressed in TNFα-treated kidney organoids are relevant to individuals with poor outcome FSGS/MCD. Together, these data demonstrate that organoid cultures can both identify molecular pathobiology involved in disease phenotype and identify subgroups of individuals within a disease for whom this pathobiology is most relevant.

To further explore the disease relevance of the ten overlap genes in Fig. 5c, we returned to the NEPTUNE cohort depicted in Fig. 5a. We interrogated snRNA-seq data[25] of a subset of ten participants for whom these data were available. This subset consisted of five participants with high TNFα activity score (cluster 3) and five participants with low TNFα activity score (cluster 1 or 2) (Fig. 5g, h, Supplementary Fig. 5C). Even when transcript expression was combined across all cell types, most of the genes demonstrated differential expression in tissue from high versus low TNF activity (Fig. 5g)[31,32]. When the relatively small differential *CXCL10* expression in Fig. 5g was further examined on a cell type specific level (Fig. 5h), more pronounced differential expression

was revealed in certain cell types in the high TNFα activity group (arrows), including immune cells as expected, as well as kidney cell types (DTL, descending thin limb), endothelial cells and fibroblasts. Similar cell-type specific expression findings could be seen in all ten overlap genes in the human kidney samples as well as TNFα-treated organoids (Supplementary Fig. 5C, D respectively). Taken together, these findings indicate that TNFα-treated kidney organoids capture key molecular mechanisms involved in poor outcome FSGS/MCD.

## TNFα-dependent molecules C3 and VCAM1 can stratify diseased kidney tissue

We further explored the ten overlap genes (shared between the 272-gene Tissue TNF signature and the 322-gene Organoid TNF signature, Fig. 5c) as potential biomarkers of TNFα pathway activation, especially those discovered in the organoid secretome (strategy shown in Fig. 5a). Two genes (*C3* and *VCAM1*) were of particular interest given previous descriptions of their potential use as biomarkers in kidney disease (Table 1). Figure 6a shows the reproducibility of C3 and VCAM1 gene expression in TNFα-treated organoids at transcription and secreted protein levels, confirming our proteomic data (Fig. 4d, f, Supplementary Data 5, 7). Single cell transcriptional profiling in Fig. 6b shows the expression of *C3* and *VCAM1* in TNFα-treated organoid kidney cell types, while expression of *C3* and *VCAM1* for all organoid cell types with and without TNFα treatment is shown in Supplementary Fig. 6A, B. Immunoblotting detected complement C3 in the media from organoid cultures only in the presence of TNFα (Supplementary Fig. 6C).

To further confirm these findings, we performed additional experiments in kidney organoids generated from: 1) a second human male iPSC line using the same protocol (Supplementary Fig. 7A) and 2) a third human female iPSC line using a suspension protocol[33] (Supplementary Fig. 7C). Transcript and secreted proteins of VCAM1, CXCL10 and C3 were increased by TNFα in these organoids (Supplementary Fig. 7B) and did so in a concentration-dependent manner (Supplementary Fig. 7D). These results demonstrate the robust nature of TNFα-induced expression of these genes in kidney organoids.

Our earlier analysis suggested that this program depends on canonical TNFRSF1A-TRADD-RIP1 interaction and signaling[34] (Supplementary Fig. 4B, C). Therefore, we treated organoids with R-7050, a small molecule blocking TNFα-dependent signaling and decreasing TNFRSF1A-TRADD-RIP1 interaction (Fig. 6c). This molecule blocked TNFα-induced C3 and VCAM1 expression, at the RNA and protein levels. Together these results demonstrate that organoid kidney cell types express these genes upon TNFα stimulation via TNFRSF1A-dependent signaling, resulting in the expression and secretion of gene products.

To determine the relevance of these findings to FSGS/MCD, we returned to the NEPTUNE cohort (Fig. 5a). As shown in Fig. 6d, transcript expression of both C3 and VCAM1 was increased in the subgroup of individuals with poorer outcomes in FSGS/MCD (cluster 3). We next posited that if these genes were potential biomarkers of TNFα activity in FSGS/MCD, the expression should be higher in kidney cell types in individuals with high TNFα activity. Indeed, single nuclear transcriptional profiling demonstrated higher transcript expression of both C3 and VCAM1 in descending thin limb (DTL) tubular cell type (Fig. 6e),

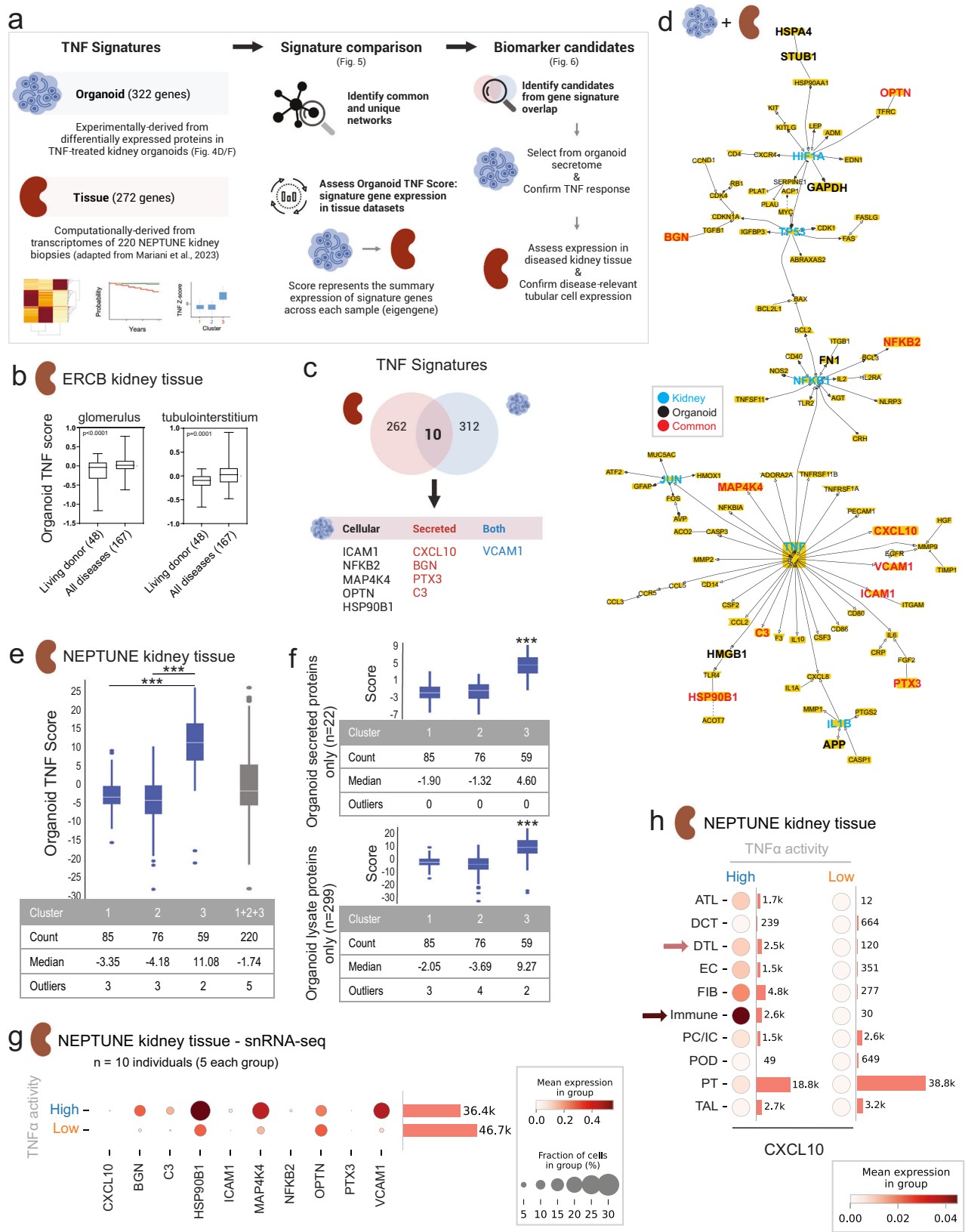

even when compared to immune cells. When these tubular cell types were isolated and separated by TNFα activity status, transcript expression of both C3 and VCAM1 was significantly higher in cells from high TNFα status individuals.

To confirm that VCAM1 protein is expressed in DTL, we performed immunofluorescence analysis of human kidney tissue. DTL was identified by AQP1-expressing cells within the medulla that did not stain with LTL (Supplementary Fig. 8A). As expected, VCAM1 protein was localized in the parietal epithelial cells (Supplementary Fig. 8B, top row) and very rarely in the proximal tubule (Supplementary Fig. 8B, bottom row). Analysis of kidney tissue from individuals with FSGS revealed VCAM1 staining in the DTL in two individuals with relatively

**Fig. 5 | Proteome-based gene signature of TNFα activation in organoids identifies group of individuals with poorer clinical outcomes in proteinuric kidney disease. a** Schematic defining origins of Organoid and Tissue TNF gene signatures (left column) as well as Organoid TNF score (bottom middle column) used in this manuscript, with special attention to FSGS/MCD cohort with the NEPTUNE study (bottom left column) demonstrating poorer outcomes and higher TNF activity for individuals in cluster 3 as in Mariani et al. (2023); right column shows strategy for identifying potential biomarker candidates using Organoid TNF signature genes. **b** Z-scores showing summary expression of organoid TNF signature genes in human kidney diseases, generated from ERCB microarray data from microdissected human kidney biopsy tissue (boxplots for the 268 in ERCB of the total 322 genes assessed, Supplementary Data 10). Plots: median, boxes 25–75% percentile, whiskers represent the min. to max. values. Numbers of individual samples are indicated in parentheses. LD = Living Donor. Unpaired *t* test. **c** Venn diagram comparing gene sets of Organoid and Tissue TNF signatures (Fig. 4d, f, Supplementary Data 10). **d** Literature-based network generated from the combination of Organoid and Tissue TNF signature genes (total 584). **e, f** box and whisker plots (white line, mean; box, 75%; whiskers, 90%) showing summary gene expression in diseased human kidney based on the Organoid TNF signature score in FSGS/MCD clusters (319 of the 322 genes detected) as in (**a**), and divided by proteins in organoid secretome (top, all 22 genes) and cell lysates (bottom, 299 of 302 genes); 2 proteins were expressed in both; ***$p < 1.85 \times 10E{-}11$. Unequal variance two-tailed *t* test. **g** Dot plots of 10 TNF signature overlap gene expression from (**c**) in individuals with FSGS/MCD separated by TNFα activity status, generated from snRNA-seq data. **h** Dot plots of *CXCL10* expression separated by diseased kidney TNFα activity status and cell type. Arrows highlight cell types of interest with higher expression in individuals with high TNFα status; cell type cluster names as in Supplementary Fig. 5C. **a, c** created using BioRender. Source data are provided as a Source Data file.

**Table 1 | Common genes between previously reported TNFα transcriptomic signature and TNFα-regulated proteins and examples for their described function in kidney disease**

| Gene | | |
|---|---|---|
| BGN | A soluble proteoglycan associated with inflammatory kidney diseases; involved in modulating inflammatory signaling through Toll-like receptors thought to originate from activated macrophages; tissue based biglycan has been detected in crescentic GN; DKD and LCDD and amyloidosis. | [67–69] |
| HSP90B1 | Part of a group of ER stress response proteins involved in immunoregulation; affects processing and transport of secreted proteins; a polymorphism of HSP90B1 has been associated with glucocorticoid response in individuals with SLE. | [70,71] |
| CXCL10 | Increased in diverse kidney diseases: MesPGN, AKI, LN, DKD; expressed by resident kidney cells at low levels (note CXC3 receptor not expressed in our organoids by RNA-seq data). | [72,73] |
| ICAM1 | Expressed in kidney tubules in IgAN, has been suggested as a marker of tubulointerstitial injury and predictor of disease progression. | [74] |
| VCAM1 | Expressed in PTECs in a variety of inflammatory kidney diseases, including diabetic kidney disease, urinary VCAM1 is suggested as biomarker of lupus nephritis activity. | [75,76] |
| C3 | Expression increases with TNFα stimulation in human glomerular endothelial cells. | [77] |
| NFKB2 | Increased nuclear localization in complex with polycystin-1 tail and STAT6 resulting in inappropriate gene expression in ADPKD. | [78] |
| MAP4K4 | Expressed in response to TNFα stimulation, reports of association with kidney disease are limited. | [79] |
| OPTN | Association with DKD, inhibition of cellular senescence through mitophagy via NLRP3 inflammasome inhibition in DKD. | [80,81] |
| PTX | Expressed in response to inflammatory stimuli and secreted into the plasma where levels inversely correlate with GFR and cardiovascular disease; expressed in proximal tubular epithelial and mesangial cells as well as kidney fibroblasts. | [82,83] |

*GN* glomerulonephritis, *DKD* diabetic kidney disease, *LCDD* light chain deposition disease, *ER* endoplasmic reticulum, *SLE* systemic lupus erythematosus, *MesPGN* mesangial proliferative glomerulonephritis, *AKI* acute kidney injury, *LN* lupus nephritis, *IgAN* IgA nephropathy, *PTECs* proximal tubular epithelial cells, *ADPKD* autosomal dominant polycystic kidney disease, *GFR* glomerular filtration rate.

low eGFR, which was not detected in DTL in two with higher eGFRs (Fig. 6f, Supplementary Fig. 8C).

## Discussion

Kidney organoids generated from human pluripotent stem cells (hPSC) have tremendous potential to advance our understanding of kidney development and disease[14,18,19,35]. However, a thorough understanding of these complex structures as well as their relevance to human kidney disease is critical to their successful implementation as a model system. Here, we describe a comprehensive, deep analysis of proteins expressed and secreted by hPSC-kidney organoids and tie these findings to transcriptional changes that occur relative to organoid culture duration, fundamental knowledge needed to advance the use of kidney organoids to model human disease. Crucially, however, we extend our study beyond solely a description of the expression of these gene products by also demonstrating the functionality and relevance of gene product networks in our organoid model system to TNFα-associated kidney disease.

We demonstrated how we could use our defined system to learn novel pathobiology relevant to TNFα associated kidney disease. TNFα itself does not cause kidney disease, but programs associated with accelerated kidney disease share similarities with TNFα controlled transcriptional programs, especially on the individual patient level[25]. In the organoids, the absence of immune cells and vasculature allowed a reductionist approach to define the TNFα-induced proteome response relevant to intrinsic nephron cell population in FSGS/MCD. First, we built on previous tissue-based findings suggesting TNFα activity was associated with poor outcomes in FSGS/MCD (Fig. 5a). The proteome of TNFα-treated organoids demonstrated a canonical response reminiscent of that observed in lipopolysaccharide (LPS) triggered kidney disease models. The TNFα response involved NFKB2 (immune response, cell development, ECM organization) and TNFα response-specific proteins such as VCAM1, ICAM1, GBP1, ARHGEF2 as well as kinases such as MAP4K4 (implicated in inflammation, programmed cell death), SLK (cell differentiation) and ILK (aging, ECM organization, cell development). Several transcription factors were enriched with TNF exposure, including EGLN1 (also known as PHD2, a hypoxia inducible factor). To further assess the impact of TNFα stimulation on organoids, we also evaluated the secretome (proteins secreted by organoids). We showed that organoids secrete more than 100 proteins, among these CXCL10, a protein active in many autoimmune diseases[36] and papilin, an extracellular matrix glycoprotein, with unknown kidney relevance[37]. Detection of other proteins known to be associated with kidney disease (including complement proteins, proinflammatory extracellular matrix, and cytokines) was also enhanced in TNFα-treated organoid proteome.

In total, our proteomics analysis revealed 322 proteins differentially expressed and/or secreted by TNFα-treated organoids. We showed that summary gene expression from this protein set was significantly higher in a subgroup of individuals with poorer outcomes in FSGS/MCD. Although on a gene set level, the organoid-based signature (322 proteins) demonstrated limited overlap with the tissue-based

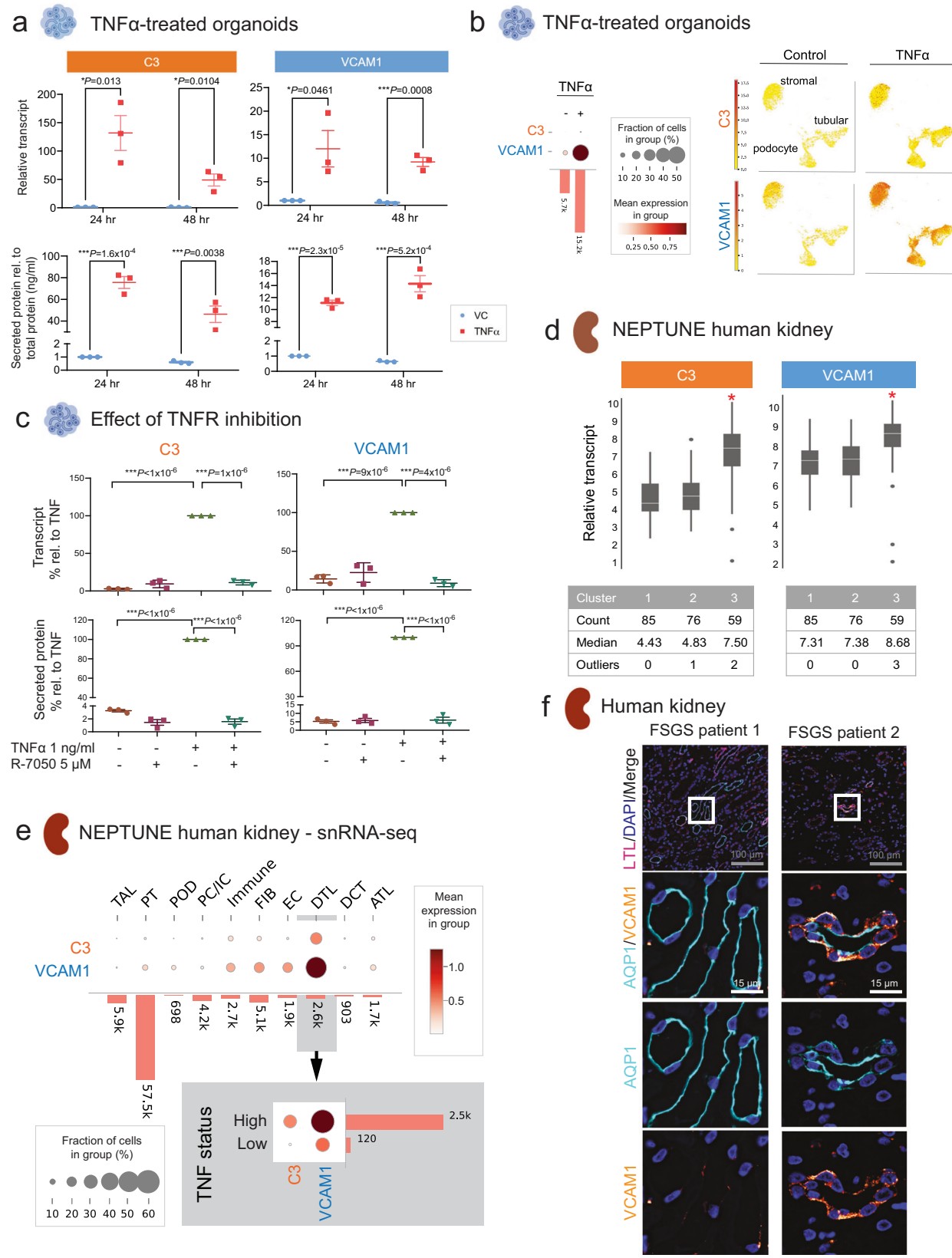

signature (272 genes), we attribute this to the focused interrogation of a limited number of cell types in organoids relative to the more heterogeneous cell population of human kidney biopsy tissue, providing a more focused kidney cell-based assessment of protein expression. Moreover, the seemingly disparate gene sets importantly converge on a TNFα-centric network, reinforcing the relevance of organoids to

FSGS/MCD. Identification of the ten common genes shared between both signatures creates opportunities to: (1) prioritize investigations of molecular pathomechanisms central to FSGS/MCD (2) identify potential biomarkers for the 272 genes identified in the human kidney signature, as well as (3) align response elements in organoid modeling with human disease for ex vivo testing of potential therapeutics.

**Fig. 6 | Organoid omics data identify potential biomarkers of kidney disease.**
**a** Expression of C3 and VCAM1 from Fig. 5c increased in D25 kidney organoids treated with TNFα for 24 h and 48 h, as measured in cell lysates by qRT-PCR (top) and in organoid culture supernatants by ELISA (bottom). Means of 3 separate experiments indicated by bold horizontal lines, with SEM error bars. Unpaired *t* test. **b** Dot plots and UMAPs of *C3* and *VCAM1* expression in kidney cell types from scRNA-seq analysis of TNFα-treated D24 kidney organoids (Fig. 4a). See also, Supplementary Fig. 6A, B. **c** Expression of *C3* and *VCAM1* in kidney organoids co-treated with TNF receptor inhibitor, R-7050, starting 1 h prior to treatment with TNFα for 24 h, as measured in cell lysates by qRT-PCR (top) or in organoid culture supernatants by ELISA (bottom). Results were reported as a percentage relative to TNFα alone. Data are represented as mean ± SEM of 3 independent experiments. Unpaired *t* test. **d** Mean *C3* and *VCAM1* expression in the same bulk transcriptional profile clusters from diseased human kidney tissues as in Fig. 5e. *$*p < 0.01$. Unequal

variance two-tailed *t* test. **e** Top: dot plots of *C3* and *VCAM1* expression by cell type in diseased human kidney generated from snRNA-seq analysis (Fig. 5g, h). Bottom: dot plots of the cell type cluster with the highest *C3* and *VCAM1* expression (DTL – distal thin limb) separated by TNFα status as in Fig. 5g, h. Cell cluster names as in Supplementary Fig. 5C. **f** Immunofluorescence imaging of sectioned human kidney biopsies of patients with FSGS showing expression of VCAM1 in the descending thin limb (DTL) compartment in patient 2 (eGFR = 39 mL/min/1.73 m²) relative to patient 1 (eGFR = 102 mL/min/1.73 m²) where no VCAM1 expression was observed. DTL segments were defined as AQP1+/LTL- tubules within the medullary region (Supplementary Fig. 8). DAPI, nuclear stain. Staining conditions were optimized in human nephrectomy tissue. 10 patient samples were stained once due to limited human biobank samples. Representative images of four patients are shown here and in Supplementary Fig. 8. Source data are provided as a Source Data file.

Additionally, half of these proteins were secreted, raising the potential for identification of additional serum or urine biomarkers of TNFα activity in human kidney disease.

To this point, we further explored two proteins identified in the TNFα-treated organoid secretome, C3 and VCAM1. We found enhanced expression of these genes in tubular cells from individuals with high TNFα signatures (by snRNA-seq), as would be expected of a kidney tissue-derived biomarker. C3 was part of a robust, orchestrated complement secretion response, which included classical cascade components C1q, C1s and C3. Both C1 complex and C3 have been suggested to control aging, apoptosis regulation and cell differentiation[38]. Kidney tubule cells produce C3 mRNA in injury[39] and express proteins C1s and C1r in tubules and glomeruli[40–42]. VCAM-1 is a marker of proximal tubule cells that transition to more regenerative cell types upon injury[43], and negatively associated with glomerular filtration rate in diabetic kidney disease (DKD)[44] and lupus nephritis[45]. Whether C3 and VCAM1 are causative or reliable markers of disease progression is yet to be determined. Further investigation into what the remaining 312 genes of the organoid TNFα signature could reveal about pathomechanisms of FSGS/MCD is also of interest.

These findings support the use of organoids to model complex human kidney diseases. Firstly, many monogenetic kidney disease markers map to the organoid proteome (Fig. 7a). These are chiefly focused on the FSGS/Nephrotic syndrome spectrum and were strongly represented in the data set, covering about 60-65% of OMIM-mentioned FSGS-associated genes, followed by genes responsible for ciliopathies (nephronophthisis, Joubert, Bardet-Biedel Syndrome), and autosomal dominant polycystic kidney disease (ADPKD) and autosomal recessive polycystic kidney disease (ARPKD), namely GANAB3 and PKHD1 in addition to PKD1, PKD2[26,46]. Organoids also expressed proteins involved in congenital anomalies of the kidney and urinary tract (CAKUT), proteins related to metabolic diseases such as Fanconi syndrome (using the expressed HNF4A nuclear hormone receptor), several metabolic storage diseases (M. Fabry, Galactosemia, GM1-Gangliosidosis, McArdle disease, Niemann-Pick disease), as well as proteins related to cystinosis[47] immunodeficiency, porphyria and hyperoxaluria. The presence of these proteins supports the development of personalized disease models of kidney phenotypes[7,48]. Secondly, more complex changes in the extracellular matrix are modeled in the organoids, enabling comparisons to multiple diseases in which fibrosis is a prominent feature (Fig. 7b, c). Indeed, our study adds to recent analyses of the glomerular basement membrane (GBM) composition of glomeruli of kidney organoids[6,16]. Thirdly, protein-level expression of multiple signaling receptors (Supplementary Data 13) facilitates further investigations into pathomechanisms as well as therapeutics. However, it is important to note that kidney organoids dramatically alter their proteome over several days' duration in culture (Fig. 1), which could greatly impact disease modeling, emphasizing the need for consistent experimentation. To provide insights in the organoid data acquired here,

together with other datasets of glomerular diseases, we made these available through an online application (https://kidneyapp.shinyapps.io/kidneyorganoids/) (Supplementary Fig. 9).

This study has limitations that are inherent to current organoid models and reflected in the omics data: (1) gene expression by immature and "off-target" cell types, (2) lack of a functional filtration barrier and collecting system due to lack of perfusion and (3) variability of organoids across cell lines, differentiation protocols and individual experiments. First, samples are biased by gene expression of immature and off-target cell types, including in response to inflammatory stimuli. As we have shown here, single-cell transcriptional profiling can complement proteome data and partly mitigate this effect. Second, the lack of functional filtration barrier precludes complete modeling of podocyte disease phenotype, such as proteinuria or podocyte effacement. Nevertheless, decreased expression of podocyte marker proteins was observed on single-cell transcript level, consistent with previous studies demonstrating change in nephrin localization with disease[7]. Moreover, in some cases the lack of perfusion can be beneficial; establishing expression by kidney cell types for some proteins (such as C3) can readily be evaluated in the organoid system without systemic interference. Third, organoid consistency is known to vary significantly across cell lines, differentiation protocols and individual experiments (batch-to-batch variations). Accordingly, we analyzed samples from at least two independent experiments and multiple biological replicates per experiment. For TNFα experiments, we included data from organoids generated from multiple hPSC lines plus a second differentiation protocol[33]; these data confirmed expression of key markers of the proinflammatory state with relevance to human tissue (Supplementary Fig. 7). Thus, management of known limitations to kidney organoid models can provide meaningful and disease-relevant data.

In conclusion, integrating omics data derived from organoid disease models and other preclinical models with data from humans with disease will drive better kidney outcomes for patients (Fig. 7d). In this study, we demonstrated that hPSC-kidney organoids can be manipulated to capture critical elements of more complex and heterogeneous responses of human kidney disease. Our omic map shows that both inflammation and (related) fibrosis can be rapidly triggered in organoids even in the absence of immune cells. These elements open the door to modeling more complex disease where multiple insults may lead to cumulative damage associated with chronic kidney disease. We suggest that other kidney disease researchers can integrate our datasets with their own to improve and refine the generation of kidney disease-relevant organoids as the technology matures further towards personalization of disease modeling and therapeutic screening.

## Methods
### Organoid culture, TNFα treatment and sample preparation
UM77-2 human (female) embryonic stem cells (NIH registration #0278, sourced from MStem Cell Laboratory) and NEPTUNE human

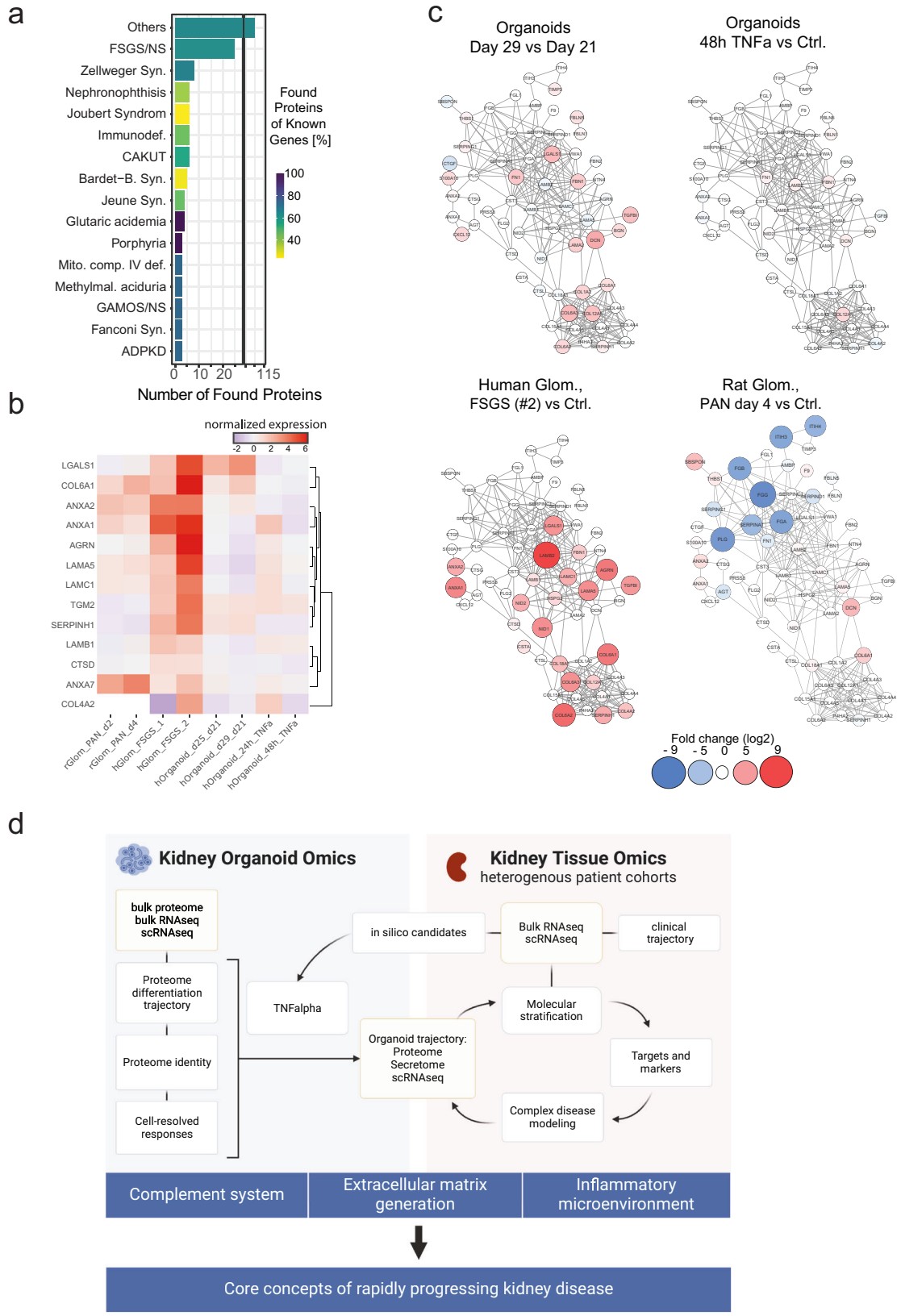

(male) iPSC line 19A (HUM00158219, HUM00120448, reprogrammed at the University of Michigan's Human Stem Cell & Gene Editing Core) were cultured in accordance with University of Michigan's Human Pluripotent Stem Cell Research Oversight Committee and NIH regulations. Though no comparison data were available for UM77-2 line, CNV assessment by Illumina bead array (Infinium

CoreExome-2.4) at UM's Advanced Genomic Core was performed in 2022 and confirmed normal karyotyping and sex. The NEPTUNE 19A line was freshly reprogrammed. Kidney organoids were generated as follows: hPSCs were dissociated with Accutase (StemCell Technologies, cat#7920) and plated onto microwell plates pre-coated with GelTrex in mTeSR1 (StemCell Technologies, cat#85850)

**Fig. 7 | Omics-expanded organoid modeling potential of kidney disease.**
**a** Number of proteins associated with monogenic kidney disease expressed in kidney organoids. **b** Heatmap of protein expression of genes associated with nephrotic syndrome expressed during organoid differentiation, D21 to D29 (visualization using Kidney Disease Explorer https://kidneyapp.shinyapps.io/kidneyorganoids/). **c** Protein–protein interaction networks created with STRING database (V.11.0) and Cytoscape (V 3.8.2), demonstrating differences in extracellular matrix protein expression in aging (upper left) and TNFα-treated (upper

right) organoids. As comparisons, corresponding networks of human FSGS tissue vs. control (lower left) as well as puromycin aminonucleoside (PAN)-treated rat kidney tissue vs. control (lower right) are illustrated. **d** Schematic of how organoids can contribute to our understanding of kidney disease through the integration of pre-clinical models with human data to drive better outcomes for individuals with kidney disease. **d** created using BioRender. Source data are provided as a Source Data file.

supplemented with 10 μM Rock inhibitor Y27632 (Tocris, cat#1254). Media was replaced with mTeSR1 plus 1.5% GelTrex at 16 h and then with Advanced RPMI (Gibco, cat#12633020) supplemented with Glutamax (Gibco, cat#35050061), 12 μM CHIR99021 (Reprocell, cat#04000402) and 10 ng/ml Noggin (R&D systems, cat# 6057NG) at 60 h. RB media [Advanced RPMI + Glutamax + B27 Supplement (Gibco, cat#17504044)] was used at 96 h, changed two days later and every 3 days thereafter. Cells were routinely monitored and tested negative for Mycoplasma infection (ATCC, cat#50-189-644FP). Organoid cultures (whole wells of cells, isolated spheroids and/or overlying culture medium) were collected at indicated time points: days 21–29 from seeding. For TNFα experiments, almost all samples were collected on day 25 following treatment with 5 ng/ml recombinant human TNFα (R&D Systems, cat#10291TA050) reconstituted in DPBS (Gibco, cat#14190144; vehicle control) for 24 or 48 h (medium replenished every 24 h) before collection. Samples collected for scRNA-seq were collected on day 24 following 24 h TNFα treatment. For TNF receptor inhibitor experiments, D24 kidney organoids were pre-treated with TNF receptor inhibitor, R-7050 (Sigma, cat#654257), at 5 μm for 1 h prior to treatment with TNFα at 1 ng/ml for 24 h. Then, samples were collected on day 25 for either RNA extraction (whole well cell lysates) or ELISA analysis (organoid culture supernatants). For proteomics studies, isolated organoids (20 organoids/sample in triplicate) were harvested and washed with ice-cold PBS, and organoid culture medium was collected and snap frozen. For ELISA studies, whole well lysates were prepared by washing the organoids in ice-cold PBS and scraping into Cell Lysis Buffer 2 (R&D Systems, cat#895347) supplemented with Halt Protease and Phosphatase inhibitor Cocktail (Thermo, cat#78440) and medium from corresponding wells was collected and snap frozen. For qRT-PCR/Bulk RNA-seq, whole wells or isolated spheroids were washed, scraped into ice-cold PBS, pelleted and lysed in 500 μL of TRIzol (Invitrogen, cat#15596026) and then total RNA was extracted using Directzol RNA Mini Prep Plus kit (Zymo Research, cat#R2072), as per manufacturer's instructions.

**Organoid immunofluorescence**
Organoid spheroids were fixed in 4% paraformaldehyde (Electron Microscopy Sciences, cat#15710), subjected to a sequential gradient of sucrose, and embedded in 20% sucrose/OCT at ratio 2:1 (Tissue-Plus, ThermoFisher Scientific, cat#4585). 5 μm cryosections were rehydrated and blocked with 5% normal donkey serum (Jackson ImmunoResearch Laboratories, cat#017-000-121) in PBS supplemented with 0.1% Triton X-100 (IBI Scientific, cat#IB07100). Slides were immunostained with primary antibodies in 3% BSA (Fraction V, Gibco, cat#15260-037) followed by appropriate Alexa Fluor secondary antibodies (Invitrogen) and mounted using Prolong Gold with DAPI (Invitrogen, cat#P36935). Samples were imaged using a Nikon A1 High Sensitivity Confocal Microscope at the University of Michigan's Microscopy Core and processed with Nikon Elements software. Primary antibodies: N-Cadherin (R&D, cat#AF6426, 1:1000); ACTA2 (R&D clone 1A4, cat#MAB1420-SP, 1:50); PDGFRA (BD Biosciences clone αR1, cat#556001, 1:200); Synaptopodin (Progen clone G1D4, cat#690094S, 1:80); NPHS1 (R&D, cat#AF4269, 1:500); TNFARSF1A (R&D clone 16803, cat#MAB225SP, 1:20); VCAM1 (Invitrogen clone 1.4C3, cat#MA5-11447, 1:50).

**Organoid ELISA**
Protein levels in kidney organoid culture media and lysates were measured using commercially available ELISA kits for CXLC10 (R&D Systems, cat#DIP100), C3 (Abcam, cat#ab108823) and VCAM1 (R&D, cat#DVC00). Samples were processed in duplicate following the manufacturer's protocol. Values were normalized to total protein content using Pierce BCA protein assay (Thermo Scientific, cat#23227). At least three independent experiments were performed and plotted using GraphPad Prism software. Statistical significance was calculated using unpaired Student's *t* test.

**Organoid qRT-PCR**
Total RNA was collected from whole well organoid cultures. One μg of total RNA was reverse-transcribed into cDNA using SuperScript First-Strand Synthesis kit (Invitrogen, cat#11904018) per manufacturer's protocol. Quantitative real-time PCR (qRT-PCR) was performed using TaqMan Fast Universal PCR Master Mix (2X) (Applied Biosystems, cat#4352042) in a QuantStudio 7 Flex Real-Time PCR System (ThermoFisher) using TaqMan Pre-Developed Assay Reagents (PDARs) as follows: C3, Hs00163811_m1; CXCL10, Hs00171042_m1; GAPDH, Hs03929097_g1; VCAM1, Hs01003372_m1. The ΔΔCq method[49] was applied to calculate the relative quantity (RQ) of target gene after normalization to GAPDH. Samples were assayed in duplicate. Graphs of at least three independent experiments were plotted using GraphPad Prism software. Statistical significance was calculated using unpaired Student's *t* test.

**Organoid bulk RNA-seq and bioinformatic analysis**
Total RNA from isolated organoid spheroids was prepared as detailed above. Library preparation using NEBNext Ultra II kit and sequencing using paired end read length of 150 bases on a NovaSeq4000 were performed by the University of Michigan Advanced Genomics Core. Fastq read quality was determined using FastQC (http://www.bioinformatics.babraham.ac.uk/projects/fastqc/), and reads aligned to the reference (ENSEMBL GRCh38.104) using STAR 2.7.8a[50]. Uniquely mapped reads were inspected for unusual distribution across known annotated features using Picard Tools (https://broadinstitute.github.io/picard/). Gene level read counts were generated using HTSeq (version0.12.4)[51] and normalized with voom[52]. PCA and hierarchical clustering were used to identify and remove samples with abnormal expression profiles due to technical issues, and the mapping statistics obtained from STAR. Data available at NCBI GEO accession number GSE213972.

**Organoid scRNA-seq and bioinformatic analysis**
Whole well organoid cultures were collected by scraping cells into ice-cold DPBS and dissociated with cold activate protease (Sigma, cat# P5380). Single cell suspensions were then submitted to the Advanced Genomics Core at the University of Michigan for library preparation and sequencing on a 10x Genomics Chromium System. Organoid scRNA-seq data processing was performed using Seurat 4.0[53]. Cells expressing >500 genes were included in the analysis. The processing steps include log transformation, scaling or linear transformation using default settings, highly variable gene identification, dimensionality reduction using principal component analysis (PCA) and Uniform Manifold Approximation and Projection (UMAP), batch correction

using harmony function embedded in Seurat and unsupervised clustering at 0.25 resolution. Data available at NCBI GEO accession number GSE213972.

## Proteomics sample preparation, organoid time course

Cell pellets (20 isolated organoid spheroids/sample in triplicate) from 2 independent experiments were lysed using urea buffer containing urea (8 M) and ammonium bicarbonate (100 mM) supplemented with 1X PPI (Thermo Scientific Halt Protease and Phosphatase Inhibitor Cocktail, cat#78440). Protein lysates were sonicated for 30 s on 10% power. After centrifugation at 220 x g for 30 min at 4 °C, the supernatant was transferred to a new tube. Protein concentrations were measured using a commercial BCA kit (Thermo Scientific). The samples were incubated with dithiothreitol (10 mM) followed by iodacetamide (40 mM) for 1 h at room temperature for the reduction and alkylation of disulfide bonds. Protein from each sample was digested with trypsin using a 1:100 ratio (1 μg trypsin per 100 μg protein). Digestion was stopped the next day by acidifying to pH 2-3 using formic acid.

## Proteomics analysis, organoid time course

For time-course analysis of organoid cell pellets, peptides were purified using in-house made stage-tips. For nLC-MS/MS analysis of proteomic data, we used a Q Exactive Plus (Thermo) instrument coupled to a nLC, with a 2.5-h gradient. A binary buffer system with buffer A: 0.1% formic acid (FA) and Buffer B 80% Acetonitrile (ACN) was used. The flow rate was 250 nl/min. The gradient settings were as follows $t = 0$ min; 4% (Buffer B), 05 min, 6%; 125 min, 23%; 132 min, 54%; 138 min, 85%; 143 min, 85% and 145 min 5%. The flow rate was constant with 250 nL/min. The Q Exactive Plus was operated in positive ion mode. One survey scan (resolution = 70000, m/z 300–1750) was followed by up to 10 MS2 scans (resolution = 17500, m/z 200–2000). Dynamic exclusion was enabled (20 s). AGC target was 3e6 for MS1 scans, and 5e5 for MS2 scans. MS data was processed using MaxQuant as detailed below. The mass spectrometry proteomics data have been deposited to the ProteomeXchange Consortium via the PRIDE[54] partner repository.

## Integration of organoid time course proteome and transcriptome

The proteomic data analysis was performed using the Perseus software suite[55] and R[56] for differential expression analysis, GO-term analysis and plotting heatmaps. For trajectory analysis, raw LFQ data were log$_2$ transformed. Missing data was accepted at a threshold of 33% across samples, with subsequent imputation of missing data. Imputation was carried out sample wise with a width of 0.3 SD and a downshift of 1.8 SD as implemented in the Perseus software. Groups were compared using two-tailed $t$ tests adjusted for multiple comparisons (permutation-based FDR 0.05), as implemented in the Perseus software. Heatmaps and clustering were carried out in Perseus or in R software via the complexHeatmap package[32] using mean-subtracted data and maximum distance clustering. GO-term annotation of proteins and GO-term enrichment of the heatmap clusters was carried out in Perseus and enrichment assessed by Fisher's exact test. Volcano plots for relevant comparisons were also generated in Perseus. Correlation analysis between protein and bulk RNA transcript was carried out in Perseus. 2D GO enrichment was also carried out using bulk RNA-seq and bulk proteomics data, with enrichment terms plotted against each other[33]. For integration with scRNA-seq, cell-type marker genes from organoid scRNA-seq were used to map protein dynamics indicative of cell type.

## Proteomic analysis of TNFα-treated organoids

Cell pellets (20 isolated organoid spheroids /sample in triplicate) from 2 independent experiments were lysed using 1:1 4% SDS/0.1 M HEPES

pH 7.4/5 mM EDTA, complemented with protease inhibitor cocktail (Roche) and denaturation at 95 °C for 5 min. 10 mM Tris(2-carboxyethyl)phosphine (TCEP) and 50 mM chloroacetamide (CAA) were used for reduction/alkylation of the samples. 50 μg aliquots were purified with paramagnetic, mixed 1:1 hydrophobic:hydrophilic SP3 beads[57]. Purified proteins were resuspended in 50 mM HEPES, pH 7.4 and digested overnight at 37 °C with trypsin (Serva, Cat#37286) in a 1:100 (w/w) ratio. Samples were acidified with 2% formic acid and peptides were purified using in-house made stage-tips. Peptides were separated on an Ultimate3000 RSLC nanoHPLC coupled on-line to an Exploris480 orbitrap tandem mass spectrometer (Thermo). The HPLC was operated in a two-column setup with an Acclaim 5 mm C18 cartridge pre-column (Thermo) and an Ionopticks aurora 25 cm column with integrated emitter tip. Separation was performed at 400 nL/min in a heated column oven at 50 °C (Sonation) with the following gradient of solvents A (H$_2$O + 0.1% FA) and B (ACN + 0.1% FA): 120 min from 2-30% B and a high-organic washout at 90% B for 9 min followed by a re-equilibration to the starting conditions (2% B). The mass spectrometer was operated with the FAIMS device at standard resolution with a total carrier glas flow of 3.8 L/min at three CVs: −40, −55, and −75V. The Orbitrap resolution for the MS1 full scan was set to 120k, whereas the MS2 scans were recorded with 1.5 s cycle time for −40V CV and 0.75 s cycle time for −55/−70V FAIMS CVs at an orbitrap resolution of 15k. Dynamic exclusion mode was set to custom with a 40 s exclusion window and a mass tolerance of 10 ppm each.

To assess the organoid secretome, proteins within the overlying cull culture medium of organoid spheroids were denatured using 1:1 4% SDS/0.1 M HEPES pH 7.4/5 mM EDTA, complemented with protease inhibitor cocktail (Roche) and heating at 95 °C for 5 min. 10 mM TCEP and 50 mM CAA were used for reduction/alkylation of the samples. 50 μg aliquots were purified with paramagnetic, mixed 1:1 hydrophobic:hydrophilic SP3 beads[35]. Purified proteins were resuspended in 50 mM HEPES, pH 7.4 and digested overnight at 37 °C with trypsin (Serva) in a 1:100 (w/w) ratio. Samples were acidified with 2% formic acid. All peptides were purified using in-house made stage-tips.

## Organoid proteome data processing

For organoid trajectory samples, raw files were searched, quantified and normalized using the MaxQuant[30] version 1.5.3.8 (FDR = 1%). Label-free quantification[30], intensity based absolute quantification (iBAQ) with log fit, and the match-between-runs feature were enabled. We used the UniProt human reference proteome as database (downloaded in January 2017) and default settings for orbitraps. Enzyme specificity was set to Trypsin/P, cysteine carbamidomethylation was set as a fixed modification (+57.021464) and methionine oxidation (+15.994914) as well as protein N-terminal acetylation (+42.010565) were set as variable modifications. Data analysis was performed using Perseus software suite (V1.5.1.6). For organoid TNFα experimental samples, raw FAIMS data were converted into MzXML files with the FAIMS_MzXML_Generator tool (v1.0.7639, ref. 36) and queried with MaxQuant v 1.6.7.0 (FDR = 1%, match between runs = on) using the UniProt reference proteome database for human (May 2020, canonical only, 20600 entries) and default settings for orbitrap instruments. Enzyme specificity was set to Trypsin/P, cysteine carbamidomethylation was set as a fixed modification (+57.021464) and methionine oxidation (+15.994914) as well as protein N-terminal acetylation (+42.010565) were set as variable modifications. The match-between-runs feature was activated with default settings.

## Organoid proteome data analysis

The data analysis was performed using the Perseus software suite and R for GO-term analysis. For organoid trajectory sample analysis and for GO-term annotation, raw LFQ data were log$_2$ transformed, triplicates were averaged, and proteins detected on all four days of differentiation were used. For volcano plotting and $t$ tests individual

samples were used with missing data accepted at a threshold of 33% across samples, with subsequent imputation of missing data. Imputation was carried out sample wise with a width of 0.3 SD and a downshift of 1.8 SD as implemented in the Perseus software. GO terms were annotated and between group testing was carried out using Student's $t$ test and Volcano plots for relevant comparisons. For organoid TNFα experimental samples, data were $\log_2$ transformed and GO-terms annotated, data were filtered to only include proteins for which we had complete data in 3 of 3 replicates within at least one group with subsequent imputation of missing data. Imputation was carried out sample wise with a width of 0.3 SD and a downshift of 1.8 SD. The FDR for Volcano plots was held at 0.05 for the trajectory analysis, at 0.1 for TNFα treated organoids. For GO term enrichment we used Fisher Exact testing implemented in Perseus (FDR 0.05) or the Enrichr R package (adjusted $p$-value < 0.05 as significant). Top GO terms per annotation term category were then used for plotting. Heatmaps and clustering were carried out in Perseus using mean-subtracted data and maximum distance clustering or using the R package complexHeatmap[32]. Venn diagrams were generated using gene symbols of the relevant data sets and produced using the Rvennr R package. GO-term annotation of intersecting and unique Venn sections was performed using EnrichR. 2D scattering of RNA count vs protein IBAQ and UniProt keyword ([www.uniprot.org](www.uniprot.org)) analysis was carried out in Perseus software with visualization of 2xSD in R. UMAP and marker lists of scRNA-seq were produced using the Seurat R package. Matrisome networks were generated from proteins in our dataset annotated with matrisome as a UniProt keyword. A list of matrisome associated identifiers was uploaded to string-db.org (V.11.0) with standard settings and 'highest confidence' selected for the minimum required interaction score. The STRING database[58] was used to generate the network which was then imported into Cytoscape (V 3.8.2)[59] and $\log_2$ fold changes mapped to the network indicating magnitude of the fold change through size and directionality through color. To assess to which degree organoids express disease-relevant proteins, we mapped proteins between OMIM disease genes and organoids.

### Human podocyte culture
Human podocytes[60] were cultured in dishes as previously described and regularly tested for mycoplasma using a commercial kit (Look-Out, Sigma). 50,000 cells were seeded in 6-well dishes (Thermo Scientific) and cultured at 32 °C with 5% $CO_2$ in RPMI 1640 (Gibco, Thermo Scientific) containing 10% FBS (Gibco, Thermo Scientific), 1% Penicillin-Streptomycin (Gibco, Thermo Scientific), 1% insulin-transferrin-sodium selenite (Thermo Scientific), 1% MEM (Gibco, Thermo Scientific), 1 mM Sodiumpyruvate (Gibco, Thermo Scientific) and 20 mM HEPES (Gibco, Thermo Scientific). 24 h after seeding, cells were washed once with PBS and medium was replaced with FBS-free medium containing 5 ng/mL TNFα (R&D Systems, same vendor as for organoid treatment) or vehicle control. After 24 or 48 h, culture medium was removed and snap frozen. Cells were washed twice with PBS and scraped into ice-cold urea buffer (8 M urea, 100 mM ammonium bicarbonate, 1X PPI), snap frozen and stored until analysis.

### Analysis of the human cultured podocyte proteome
Cells and supernatants were analyzed using MaxQuant and Perseus. Raw files were searched, quantified, and normalized using MaxQuant version 1.6.17.0 with default settings for orbitraps. The match between runs (MBR), LFQ, IBAQ and classical normalization features were enabled. We used the UniProt human reference proteome as database (UP000005640_9606, downloaded in April 2021 with 20612 entries with enzyme specificity set to Trypsin/P, cysteine carbamidomethylation as a fixed modification (+57.021464) and methionine oxidation (+15.994914) as well as protein N-terminal acetylation (+42.010565)

were as variable modifications. Data analysis was performed using Perseus software suite (V1.6.2.3). TNFα-treated podocyte data were $\log_2$ transformed and filtered to only include proteins that were measured in at least 4 out of 6 replicates in at least one group. Missing data was imputed sample wise from a normal distribution with a width of 0.3 SD and a downshift of 1.8 SD. Volcano plots were created according to a two-sided $t$ test with an FDR of 0.2 for supernatants and an FDR of 0.1 for cells.

### Human glomeruli and tubule proteomics data
Datasets were retrieved for human microdissected proximal tubules and single human glomeruli from adult human kidneys[19] to compare with our organoid data.

### Deep proteomic analysis of human glomeruli
For deep mass spectrometry analysis sieved human glomeruli were used as previously described[61]. Proteins from glomeruli were extracted, solubilized and trypsinized as detailed above. The tryptic digests were fractionated using in-house made stage-tips, applying high pH reverse phase fractionation with fresh 100 mM ammonium formate, pH 10 and stepwise increase of ACN from 0-50% (n = 8 fractions). The eight fractions were analyzed on the same Q Exactive Plus instrument as indicated above with analogous settings. Proteomics data from single tubule and glomeruli was obtained from a previous study[28].

### NEPTUNE bulk RNA-seq and snRNA-seq
Publicly available data from an existing study of individuals with proteinuric kidney disease were utilized for analysis[25]. The following information is provided for transparency and reference: the NEPTUNE (NCT01209000) study objectives, design and procedures were described in previous publications[62,63]; consent was obtained from individuals or parents/guardians at enrollment, and the study was approved (HUM00158219) by University of Michigan, Medical School Institutional Review Board. Renal biopsies were microdissected into glomeruli and tubulointerstitial compartments. Bulk tubulointerstitium kidney biopsy transcriptional data from 220 NEPTUNE participants with FSGS/MCD were accessed at NCBI GEO accession number GSE182380; single nuclear transcriptional data for ten NEPTUNE participants within this cohort, five with high intrarenal TNF activity and five with low TNF activity, were accessed at GSE213030.

### Organoid interactome
Cell-type specific protein expression was assigned as detailed above and as in Fig. 2d. NicheNet[64] was used to evaluate cell-type expression of genes encoding the 716 cell marker proteins significantly differentially expressed between days 29 and 21 (Supplementary Data 2, 4). Prior interaction potential values (bona fide) were calculated for potential cell-type specific ligand-target interactions, based on evidence of target engagement (expression of predicted target genes).

### Organoid derived proteomic TNF signature applied to NEPTUNE kidney tissue transcriptome data
The proteomic signatures of TNFα treated organoids were derived from the total of 322 differentially expressed proteins (DEPs) under TNFα treatment at 24 and 48 h (Supplementary Data 10) by combining cellular proteins (n = 302 DEPs) with those proteins secreted into the culture medium (n = 22 DEPs) (Supplementary Data 3, 4). As two DEPs were represented in both sets, there was a total of 322 DEPs. Peptide to ENSEMBL gene id conversions were performed using Biomart. Each of the three gene sets (cellular, secreted, combined) was used to compute an eigengene (first principal component) from each patient-derived tubulointerstitial transcriptome profile in patients with MCD or FSGS from NEPTUNE (GEO Accession GSE182380); only 319 of the 322 genes were expressed (299 in lysate, 22 in secretome including 2 in both).

Statistical significance between groups was calculated using two-tailed (unequal variance) Student's *t* test.

## Network and pathway analysis

For each of the two gene signatures, (i) from the previously characterized TNF gene set in human kidney tissue (*n* = 272 genes)[25], (ii) from the organoid TNFα proteome (*n* = 322 genes) and the combination of these two gene signatures (*n* = 584 genes), biologic literature-based networks were generated using Genomatix Pathway System (GePS) software (http://genomatix.de). In these networks, the 100 best connected genes co-cited in PubMed abstracts in the same sentence linked to a function word (most relevant genes/interactions) were represented. The TNF-centric network was generated from the 584 combined genes asking the software to include in the network the 10 overlap genes listed in Fig. 5c in the 100 best-connected genes. The top 20 pathway-based and signal transduction networks were generated from the individual and combined gene signatures using GePS (Supplementary Data 11, 12).

## Upstream regulator identification

Differentially expressed genes (DEGs) with absolute $\log_2$ fold change ≥1 were identified from bulk RNA-seq organoid datasets for day 29 versus day 21 (Supplementary Data 3, 14) and for TNFα-treated versus vehicle control at 24 h and 48 h (Supplementary Data 15). Upstream regulator analysis was performed from each dataset using Ingenuity Pathway Analysis (QIAGEN IPA). IPA generates an activation z-score as a statistical measure of the match between expected relationship direction and observed gene expression and considers z-score≥2 and ≤−2 significant. Upstream regulators with absolute z-scores ≥ 2 were reported (Supplementary Data 16, 17).

## Transcriptomic analysis of the European Renal cDNA Bank (ERCB) tissue

Previously generated microarray data from microdissected human glomeruli and tubulointerstitium sourced from individuals with kidney disease (*n* = 184) and healthy donors (*n* = 50) were used, accessed via GEO accession numbers GSE104948 and GSE104954[65]. Microarray assays captured a subset of 268 genes of the 322-gene product kidney organoid signature in the ERCB cohort. Gene expression was Z-transformed, and Z-scores were calculated as previously described[25].

## Kidney Disease Explorer Shiny application

Seventeen different data sets from the following prior publications were combined with data from this study for this application. Rinschen et al.[37] defined the cellular effects of puromycin aminonucleoside (PAN) under different circumstances (in vitro differentiated and non-differentiated, in vivo after two and four days). Höhne et al.[19] created proteomic datasets for various kidney diseases (FSGS, congenital nephrotic syndrome) based on individual kidney segments. Bartram et al. analyzed FSGS (podocyte cell line and primary renal epithelial cells from urine) due to G195D mutation in the ACTN4 gene[38]. Koehler et al. analyzed the effects of doxorubicin and LPS on podocytes[39]. The app is available at https://kidneyapp.shinyapps.io/kidneyorganoids/.

## Western blot for complement factor C3

Proteins in culture media were separated on 4–15% gradient SDS-PAGE gels (Bio-Rad, Criterion TGX gels #567-1083) and transferred to nitrocellulose membranes (Bio-Rad #170-4159) which were blocked and developed with 0.5 μg/mL polyclonal rabbit anti-human-C3c (DAKO #0368) in Tris-buffered saline, 1 mM EDTA, pH 7.4, with 1 mg human serum albumin (CSL Behring #109697) and 100 μg human IgG (CSL Behring #007815) per milliliter. The membrane was then washed, incubated with horseradish peroxidase-conjugated goat anti-rabbit IgG antibody (DAKO #P0448) and developed with SuperSignal West Dura extended-duration substrate (Pierce). Emission was recorded by a charge-coupled device camera.

## Organoids generated in suspension

Kidney organoids were generated from human female iPSC cell line UKEi001-A (https://hpscreg.eu/cell-line/UKEi001-A; cellosaurus[66] ID-number: CVCL_A8PR) following a modified Takasato protocol[33]. All procedures were in accordance with the Code of Ethics of the World Medical Association (Declaration of Helsinki). Human dermal fibroblasts were collected by skin biopsy from control patients enrolled in the IndivuHeart Study (PV4798/28.10.2014) reviewed and approved by the ethical committee of the Board of Physicians Hamburg (Ethik-Kommission der Ärztekammer Hamburg). Patients provided written informed consent and did not receive financial compensation. To generate organoids, hiPSCs were dissociated into single cells using Accutase (Gibco), seeded onto Matrigel-coated (Corning) 6-well plates (Nunc) at a density of 12,000 cells/cm2 in E8 media (Thermo Fisher Scientific) with Y-27632 (10 μM, Biorbyt), and incubated overnight at 37 °C and 5% CO2. This hiPSC monolayer was cultured in E6 media (Thermo Fischer Scientific) supplemented with 7 μM CHIR99021 (Sigma) from day 1 to day 4, followed by 200 ng/ml FGF9 (Peprotech), 1 μg/ml heparin (Stemcell Technologies) and 1 μM CHIR99021 from days 5 to 7. To form organoids, cells were then dissociated using Trypsin (Gibco), washed with E6 and centrifuged at 200 g. The cell pellet was resuspended in Stage 1 media [E6 media containing 200 ng/mL FGF9, 1 μg/mL heparin, 1 μM CHIR99021, 0.1%PVA (Sigma), 0.1% MC (Sigma), 10 μM Y-27632] and transferred to 6-well plates pre-treated with Pluronic-F12 (Sigma) for low adhesion conditions (day 7 + 0). Cell aggregates spontaneously formed after rotating the culture dishes on an orbital shaker (Thermo Fisher Scientific) at 70 rpm incubated at 37 °C and 5% CO2 for 24 h. Medium was switched to Stage 2 (E6 media containing 200 ng/ml FGF9, 1 μg/ml heparin, 1 μM CHIR99021, 0.1% PVA, 0.1% MC) for another 4 days (7 + 1 to 7 + 4). From day 7 + 5 onwards, organoids were cultured in Stage 3 media (E6 media containing 0.1% PVA, 0.1% MC) until the end of the experiment. Organoids were stimulated with TNFα and cell pellets and media were collected and analyzed as detailed above. Experiments were done in three independent differentiations (each with three replicates), and at the organoid age of day 23-25.

## Human kidney tissue immunofluorescence

We searched the nephropathology's sample archive for FFPE-embedded kidney biopsy tissues with the diagnosis primary FSGS. Every written biopsy report was assessed by an experienced nephrologist. Only biopsies with the distinct diagnosis "primary FSGS" with medullary components were selected for staining. Samples that showed signs of relevant comorbidities (e.g. IgAN), signs of secondary FSGS (e.g. incomplete foot-process effacement) or minimal change glomerulopathy (e.g. unaltered appearance in light microscopy) were excluded. Anonymized residual biopsy samples were exempt from ethics vote according to applicable law (HmbKHG §12). Paraffin sections were cut at a thickness of 1–2 μm and mounted on SuperFrostTM Plus microscope slides. Next, the tissue samples were immersed in xylene 3x for 10 min each, followed by a descending concentration (3×100%, 2×70%, 1×50%) ethanol series for 5 min each. Samples were immersed 3x for 5 min each in double deionized water. Target retrieval was performed using the Agilent DAKO Target Retrieval Solution pH9 (Catalog No.: S236884-2) in a Braun Multiquick FS20 steamer for 15 min, followed by a cool-down to room temperature of 45 min. The sections were then incubated in Agilent Wash Buffer Solution (Catalog N.: K800721-2) for 30 min at room temperature. Samples were incubated with primary antibodies according to vendor's guidelines in Agilent Antibody Diluent Solution (Catalog N°: K800621-2). During incubation with primary antibodies, we labeled the brush border of proximal tubules using a biotinylated lotus tetragonolobus lectin (LTL,

Vector Laboratories B-1325-2; biotinylated; 1:400). After overnight incubation at 4 °C, primary antibodies and LTL were washed off with Agilent Wash Buffer Solution 3x for 5 min. Then, sections were incubated with appropriate secondary antibodies according to vendor's guidelines, with DAPI (Sigma Aldrich D9542) as a nuclear co-staining (final concentration of 1 μg/ml) in Agilent Antibody Diluent Solution for 1 h at room temperature. Secondary antibodies were washed 3x for 5 min with Agilent Wash Buffer Solution. After immunostaining, samples were mounted with ProLong Gold (Invitrogen P36930). Primary antibodies and their respective host species and dilutions used in this study are as follows: AQP1 (SantaCruz sc-25287, clone B-11; mouse; 1:200), VCAM1 (Abcam ab134047; clone EPR5047, rabbit; 1:200). The following secondary antibodies were used: Streptavidin Alexa Fluor 488 conjugate (Invitrogen S11223; 1:400), donkey anti-rabbit Alexa Fluor 555 conjugate (Invitrogen A31572, 1:200), donkey anti-mouse Alexa Fluor 647 conjugate (Invitrogen A31571; 1:200). LED-based widefield imaging of tissues was performed using the THUNDER Imager 3D Live Cell and 3D Cell Culture (Leica Microsystems) in combination with a ×40 objective (NA: 1.10) and a ×63 objective (NA, 1.10) after optimizing LED-intensity and exposure times. Fiji imaging software (Max Planck Institute of Molecular Cell Biology and Genetics) was used to navigate the raw files. File navigation, adjustment of color balance and image analysis were performed using ImageJ software (version 2.1.0/1.53c).

### Statistics and reproducibility
No statistical methods were used to predetermine the sample size. The experiments were not randomized. The Investigators were not blinded to allocation during experiments and outcome assessment. One set of experimental data from TNF-treated organoids was discarded after it was discovered that the TNFα response was inadequate and the rTNFα used for the experiment was past expiration.

### Reporting summary
Further information on research design is available in the Nature Portfolio Reporting Summary linked to this article.

## Data availability
Organoid bulk and single cell RNA-seq data were uploaded to NCBI GEO accession number GSE213972. A Cell x Gene visualization instance is available here: http://18.188.163.197/. Proteomics analysis, organoid time course. The mass spectrometry proteomics data have been deposited to the ProteomeXchange Consortium via the PRIDE[54] partner repository with the dataset identifiers. Project Name: LC MS/MS of kidney organoids during differentiation from day 21 to 29. Project accession: PXD029716. Proteomic analysis of TNFα-treated organoids. The mass spectrometry proteomics data have been deposited to the ProteomeXchange Consortium via the PRIDE[54] partner repository with the dataset identifiers. Project Name 1 (organoid spheroid cell lysate): Proteome analysis of kidney organoid cells during TNFα stimulation. Project accession: PXD029718. Project Name 2 (secretome): Proteomic analysis of kidney organoid supernatant during TNFα stimulation. Project accession: PXD029696. Analysis of the human cultured podocyte proteome. The mass spectrometry proteomics data have been deposited to the ProteomeXchange Consortium via the PRIDE[54] partner repository with the dataset identifiers. Project Name 1: Proteomic analysis of cultured human podocytes stimulated with TNFα - cell pellet. Project accession: PXD032107. Project Name 2: Proteomic analysis of cultured human podocytes stimulated with TNFα – supernatant. Project accession: PXD032130. NEPTUNE bulk RNA-seq tubulointerstitium kidney biopsy data from NEPTUNE participants with FSGS/MCD were accessed via GEO accession number GSE182380. Single nuclear RNA-seq data for ten NEPTUNE participants were accessed via GEO accession number GSE213030. Transcriptomic (microarray) data of the European Renal cDNA Bank (ERCB) tissue

were accessed for glomerular and tubulointerstitial compartments via NCBI GEO accession numbers GSE104948 and GSE104954. No novel code was developed for this research. Source data are provided with this paper.

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

## Acknowledgements

The authors acknowledge excellent technical support by Stefan Gatzemeier, Mette Løbner, Yuee Wang, Emily Tanner, and Annette Gudmann Hansen. We thank Julio Saez-Rodriguez for helpful discussions. The authors also extend their gratitude to investigators and participants involved in the ERCB and the NEPTUNE consortia (full members list in Supplementary Information) and data made available through funding and/or programmatic support. The Nephrotic Syndrome Study Network (NEPTUNE) is part of the Rare Diseases Clinical Research Network (RDCRN), which is led by the National Center for Advancing Translational Sciences (NCATS) through its Division of Rare Diseases Research Innovation (DRDRI) and funded under grant number U54DK083912 as a collaboration between NCATS and the National Institute of Diabetes and Digestive and Kidney Diseases (NIDDK). RDCRN consortia are supported by the RDCRN Data Management and Coordinating Center (DMCC), funded by NCATS and the National Institute of Neurological Disorders and Stroke (NINDS) under U2CTR002818. Additional funding and/or programmatic support was provided by the University of Michigan, NephCure Kidney International, the Halpin Foundation as well as the George M. O'Brien Michigan Kidney Translational Research Core Center and Applied Systems Biology Core (P30DK081943) (with support to S.E., R.M., C.C.B., F.E., B.G., F.A., P.J.M., D.F, W.J.) J.L.H. and M.K. are supported by NIH NCATS UG3-TR-003288-01 and JDRF 5-COE-2019-861-S-B. This study was in part supported by the 3R (Replace, Reduce, Refine) Start-up Funding Program, awarded by the Medical Faculty Hamburg in 2018 to F.B. and Arne Hansen. M.M.R was supported by the DFG (RI 2811/1-1 and RI 2811/2-1, FOR2743, and SFB1192-project B10), by the Young Investigator Award from the Novo Nordisk Foundation, grant number NNF19OC0056043, the Carlsberg Young Investigator fellowship as well as Aarhus universitet forskningsfond, and an endowed professorship by the Klinikum Bad Bramstedt. M.M.R. thanks the Klinikum Bad Bramstedt for support. This project was also supported by the European Union's Horizon 2020 research and innovation funding programme under the Marie Skłodowska-Curie grant agreement No 754513 and The Aarhus University Research Foundation (to M.M.R). V.G.P. was supported by DFG (CRC1192—project B09), BMBF (eMed Consortia Fibromap) and the Novo Nordisk Foundation (Young Investigator Award; NNF21OC0066381). T.B.H. was supported by DFG (CRC1192, HU 1016/8-2, HU 1016/11-1, HU 1016/12-1), BMBF (STOP-FSGS-01GM2202A, NephrESA-031L0191E, UPTAKE-01EK2105D), the Else-Kröner Fresenius Foundation (iPRIME-Clinician Scientist Program), and funding from the Innovative Medicines Initiative 2 Joint Undertaking under grant agreement No. 115974 (Beat-DKD). This Joint Undertaking receives support from the European Union's Horizon 2020 research and innovation program and EFPIA with JDRF. D.K. and B.D. received intramural funding from the UKE (Clinician Scientist Program). M.T.L. was supported by the BMBF (STOP-FSGS-01GM2202A, UPTAKE-01EK2105D). F.G. received DFG funding (DFG GR3933/1-1, SFB 1192 B 09). E.H. received funding through the Heisenberg-Program of the DFG (DFG 414280945). F.B. was supported by the Else Kröner Fresenius Stiftung (2021_EKMS.26) & the German Society of Nephrology. This study was supported by grants from the Deutsche Forschungsgemeinschaft (DFG) (INST 337/15-1, INST 337/16-1 & INST 152/837-1).

## Author contributions

All co-authors have contributed to the manuscript. J.L.H., M.M.R. conceptualized and supervised this study. M.L., J.E.S., M.M.R. wrote the original draft. M.L., J.E.S., C.C.B., S.E., M.F., V.V.W. did the formal analysis. J.E.S., C.C.B., S.E., M.F, R.M., V.V.W., W.J. developed the methodology. M.L., C.C.B., S.E., A.H., L.L.B., R.M., F.E., F.A., D.F., P.J.M., B.G. performed bioinformatic processing and analyses in the study. J.E.S., M.F., V.V.W., A.M., P.J.M. performed kidney organoid experiments (including generating and analyzing qPCR, ELISA, IF data).

S.D.L., F.B. grew and stimulated organoids (in suspension protocol). A.M.B., F.D. performed proteomics. M.L., J.E.S., S.E., M.F., A.H., L.L.B., V.V.W., P.B. contributed to data curation. B.D., L.R. performed cell culture (separate from organoids) or worked on human glomeruli isolation. D.K., V.G.P. stained and imaged human biopsy tissue. M.L., C.C.B., S.E., A.H., R.M., F.A., D.F., P.J.M., M.P.R., M.C., F.D. helped with visualization. M.L., S.E., J.L.H., M.M.R. were involved in project administration. E.M., T.W., M.N., E.H., F.G., M.T.L., T.B.H., H.S., S.T., L.H.M., M.K. provided resources. F.G., T.B.H., M.K., F.D., J.L.H., M.M.R. were involved in funding acquisition. All authors reviewed and provided valuable feedback on the manuscript.

## Funding

## Competing interests

M.K. reports grants and contracts, outside of this study, through the University of Michigan with the National Institutes of Health, Chan Zuckerberg Initiative, AstraZeneca, NovoNordisk, Eli Lilly, Gilead, Goldfinch Bio, Janssen, Boehringer-Ingelheim, Moderna, European Union Innovative Medicine Initiative, Certa, Chinook, amfAR, Angion, RenalytixAI, Travere, Regeneron, IONIS and Maze Therapeutics. He has received consulting fees through the University of Michigan from Astellas, Poxel, Janssen, and UCB. M.K. serves on the NIH-NCATS council and is on the board of NephCure Kidney International. In addition, M.K. has a patent PCT/EP2014/073413 "Biomarkers and methods for progression prediction for chronic kidney disease" licensed. T.B.H. reports having consultancy agreements with AstraZeneca, Bayer, Boehringer-Ingelheim, DaVita, Fresenius Medical Care, Novartis, and Retrophin; receiving research funding from Amicus Therapeutics, Fresenius Medical Care; and being on the editorial board of Kidney International and the advisory board of Nature Review Nephrology. L.H.M. serves on the scientific advisory board for Chinook Therapeutics, Travere Therapeutics and Calliditas Therapeutics. She has grant support from Travere Therapeutics and Boeringer-Ingelheim. The other authors have no competing interests to declare.

## Additional information

[1]III. Department of Medicine, University Medical Center Hamburg-Eppendorf (UKE), Hamburg, Germany. [2]Hamburg Center for Kidney Health (HCKH), University Medical Center Hamburg-Eppendorf, Hamburg, Germany. [3]Department of Internal Medicine, Division of Nephrology, University of Michigan Medical School, Ann Arbor, USA. [4]Department of Biomedicine, Aarhus University, Aarhus, Denmark. [5]Department of Computational Medicine and Bioinformatics, University of Michigan Medical School, Ann Arbor, MI, USA. [6]Department of Pathology, University Medical Center Hamburg-Eppendorf (UKE), Hamburg, Germany. [7]Section Mass Spectrometric Proteomics, University Medical Center Hamburg-Eppendorf (UKE), Hamburg, Germany. [8]Department of Clinical Medicine, Aarhus University, Aarhus, Denmark. [9]Department of Pathology, Aarhus University Hospital, Aarhus, Denmark. [10]Aarhus Institute of Advanced Studies (AIAS), Aarhus, Denmark. [11]These authors jointly supervised this work: Jennifer L. Harder, Markus M. Rinschen. ✉ e-mail: jlharder@med.umich.edu; m.rinschen@uke.de

