## [Peer Review File · Nature Communications]

An integrated organoid omics map extends modeling potential
of kidney diseaseREVIEWER COMMENTS

Reviewer #1 (Remarks to the Author):

Lasse M et al

Nature Communications 22-38308

An integrated organoid omics map extends modeling potential of kidney diseases

Summary

Kidney organoids, especially those derived from patient specific pluripotent stem cells, are thought to be an improved model to study kidney disease over the classical monoculture or co-culture systems.

Lasse M et al present an in-depth interrogation using primary bulk proteomic, bulk RNAseq, and sc-/sn-RNAseq transcriptomic data sets derived from organoids cultured for specific time points as well as organoids stimulated by cytokines known to be associated with progressive kidney diseases. In addition to primary data the authors integrate scRNAseq and snRNAseq datasets available through public consortia (ECRB and Neptune). The organoid model recapitulates finds expected for expression of ECM with aging and increase relative activity of stromal cells over podocyte-like cells with aging. Stimulation of organoids with TNF α increased expression of 320 gene products many that are also associated with poor outcomes in patients with proteinuric diseases/syndromes. Two proteins, complement C3 (C3) and Vascular cell adhesion protein 1 (VCAM-1) previously identified as proteinuric disease theranostic biomarkers highlighted the potential for kidney organoids use in biomarker engagement studies.

Comments & Critiques

Lasse M et al present a tour de force of studies addressing the functional relevancy of kidney organoids to the study of human proteinuric diseases. Key points in the manuscript are (1) defining the expressional programs in organoids with transitioning across the glomerular phenotype into an aged phenotype, (2) integrating the proteomic and transcriptomic datasets, (3) comparing the organoid expressional programs with RNAseq data sets for microdissected microstructures (glomerular, tubular) from patient biopsies, (4) defining organoid response to cytokine (TNF α) stimulation, (5) integrating organoid TNF α response with imputed TNF α response results from patient data, and (6) a targeted analysis of TNF α responsive proteins, C3 and VCAM1, to define patient response.

(1) Analysis of kidney organoid trajectory

Summary: Previous studies of kidney organoid development documents a shift in protein expression toward glomerular differentiation at 3wk. Therefore, the shift in proteome and transcriptome was followed at 21d and 29d. Large numbers of proteins were identified (6,700) with sufficient information for quantitative analysis on 5,400 proteins. Differential protein abundance between 21d and 29d was observed for 350 increasing and 428 decreasing proteins. The changes for marker proteins were used as surrogates to infer cell specific or compartment specific changes such as SYNPO or NPHS1 for podocytes and ACTA2 and PDGFRA for extracellular matrix compartments.

Critique: Why are ACTA2 and PDGFRA used as markers for ECM? Figure 1D documents many GO processes specific to collagens, laminins, and other proteins that would be a better phenotype for fibrotic diseases? If this has not been defined then it should be added at some contextual level. If the PDGFRA represents a unique aspect of profibrotic disease and ECM upregulated that should be acknowledged.

(2) Organoid proteome-transcriptome integration and cellular origins of proteins

Summary: Bulk transcriptomic expression changes were used to access regulation of gene expression and then correlated to protein abundance. Unsurprisingly the bulk correlation was low but consistent as measured by a word enrichment. scRNAseq analysis of organoids resulted in 14 cell specific clusters with a majority as kidney specific. Annotation of the 14 cell specific subtypes on to the proteomic data noted the largest proteome changes occurred in the stromal cell population and decreases in cells corresponding to early glomerular

epithelial cells and podocytes. To better address correlation of gene and protein signatures the authors took the top 30 differentially abundant gene products and integrated expressional changes. The majority (90%) were renal cell specific observations decreased over time in relative abundance at both the transcript and protein level. The majority of the stromal cell specific observations increased over time.

Critique: What is word enrichment? Is this a feature for analysis of GO annotation data? At this time it would appear to be a word cloud analysis? Is this the most robust analysis of GO terms? The authors data suggest the largest (top30 changes up or down in gene/protein expression) are not regulated post-transcriptionally. Is there a common feature of upstream gene activators such as TF that might inform of gene expression activation? How are these TF integrated into TNF α signaling?

(3) Comparison of organoid proteome organization with mature human tissue

Summary: To better understand the expressional programs of the organoids the authors integrated the proteomic data of the organoid with the proteomes of isolated human glomeruli, isolated human tubules, and immortalized podocytes. GO terminology for the comparisons organoid to glomerular or tubular microstructures were enriched in terms associated with receptor mediated endocytosis and ECM. Podocytes were enriched uniquely in terms associated with metabolism.

Comment. Is the integration of the podocyte data here relevant as the GO terms enriched would suggest single cells in culture are ontologically enriched in terms for metabolism and not necessarily for terms associated with higher order interactions with the 3D environment. The authors note that organoids are superior to podocytes in culture perhaps that is too passive of a statement.

(4) Organoid responds to TNF α with activation of proinflammatory proteins

Summary: Prior investigations documented the response of kidney organoids to TNF α and the correlation of differential gene expression with clinical populations for FSGS and MCD. Key targets observed in this study were CCL2/MCP-1 and TIMP1. The kidney organoids expression of TNF α receptors was observed only at the gene level and only significantly for TNFRSF1A. Immunofluorescence studies suggested TNFRSF1A expression would be specific to N-cadherin expressing (proximal tubular) cells. The organoids responded to TNF α to alter the expression of approximately 300 proteins including increased expression of 145 proteins and decreased expression of 157 proteins. Included in the increased are VCAM-1, NF- κ B2, and ITGA3; all markers of early glomerular epithelial cells.

Critique: Please address why expression of TNFRSF1A in proximal tubular cells impacted expression of glomerular epithelial cells. Also please justify the use of the secretome data. A large amount of protein was analysis using state of the art methods and the secretome analysis yields a paucity of protein identification (120 on average). Moreover, a large number of these identification are naturally low abundant and yet readily observed. How was the presence in the growth media or as ECM protein components intrinsic to the organoid culture system eliminated? The authors have to address how these data are robust and specific to the organoid and not related to experimental conditions or treatments.

(5) Proteome alterations in TNF α -induced organoids help stratify disease human kidney tissue

Summary: The authors sought to translate the organoid findings into human disease by comparing the signature of gene activation to previously characterized TNF α -associated changes in 272 genes associated with poorer outcomes in MCD and FSGS populations. The authors focused on organoid renal cell populations and sought to exclude stromal cell populations. The gene signature was expanded from 272 to 320 gene products. Within those two groups the authors focused on 10 gene products for elaboration and discussion (five of the 10 proteins were resident in the TNF α responsive secretome data). GO analysis of the 272 gene products suggests comparatively to the 320 gene products a focus on metabolism rather than ECM and extracellular cytokine signaling. Application of the 320 gene signature to the kidney tissue transcriptome cluster identified an association with a patient cluster having poor outcomes to therapy. The weak association of the organoid signature with the patient

data suggests patient data is more granular and complex. Thus perhaps the organoid data can drive subgroup analysis.

Critique: Correlation of TNF α responsive expressional programs is important for understanding outcomes in the targeted patient populations. As of now it is unclear how this process was performed. The changing between 272, 320, and 10 gene products is confusing. It appears that at some states these analyses were independent, as the 272 were completely part of the 320 or that there are only 10 gene products overlapping. Do these two panels remain independent during the analyses or were the 320 genes all-inclusive of the 272 plus some 48 (not 10) additional gene products? What is the enrichment of pathways in the entire 48 overlapping genes?

(6) TNF α -dependent molecules C3 and VCAM1 can stratify disease kidney tissue

Summary: The kidney organoid and patient RNAseq data sets overlapped in the TNF α gene response signature with 10 gene transcripts. Two transcripts of prior interest due to associations with pro-inflammatory character were complement C3 and VCAM1. Both C3 and VCAM1 transcripts and protein increased with TNF α stimulation, and C3 abundance in organoid cell culture media were increased. Examination of C3 and VCAM1 expression in snRNAseq data from human biopsy material localized C3 and VCAM1 expression to the distal thin limb tubular cell types and treatment of micro-dissected DTL tubular cells with TNF α recapitulated the organoid expression changes.

Critique: These data are intriguing as C3 and VCAM1 are relevant and known markers for renal diseases. What is a surprise is the relationship to the DTL.

Overall critique: The authors have done a masterful job addressing the value of kidney organoids as a surrogate for the study of human kidney disease. Data are presented for (A) differential expression of proteins between the 21d and the 29d program to be stromal cells and glomerular cells resulting in the enrichment in stromal cells and glomerular cells as sources of protein expression; (B) regulation of gene expression programs by TNF α the authors note the expression of TNFRSF1A by proximal tubular cells, and (C) TNF-gene response program marker components C3 and VCAM1 to map expression to the DTL. How did the age of the organoid at treatment time for (B) and (C) compare with (A). How do the authors explain the changes to the TNF α responsive program taking place in the tubular compartments and not the glomerular compartment or the stromal compartments? Is it possible to experimentally spatially resolve organoid DTL, PTEC and glomerular expression for TNFRSF1A (FISH) with C3 or VCAM1 with SYNPO or NPHS1 with ITGA3?

Reviewer #2 (Remarks to the Author):

This study is to characterize kidney organoids by proteomics and transcriptomics and validate the organoid model to study TNF α -mediated renal disease with data from NEPTUNE study. Overall, the manuscript contains nice data sets that would be helpful for the research communities. However, lack of interventional/mechanistic studies on TNF α -induced podocyte diseases weakened the scientific rigor and impact of the study. The followings are my suggestions for improvement.

Major:

1. Batch-to-batch variations are one major challenge in organoid research. Only one line of human pluripotent stem cells is used in this study, and it is unclear how many independent experiments were performed.
2. There are several differentiation protocols for kidney organoids, yet only one protocol is used in this study. The rationale for the selection of the differentiation protocol is unclear.

Differentiation protocols that do not follow the metanephric kidney development resulted in poor differentiation and fibrosis development in short-term culture (e.g. PMID: 30033089). Indeed, fibrosis is induced in 4 weeks of differentiation in this study as shown in figure 1.

3. In the line 174, I believe authors meant E-cadherin (CDH1). CDH1 is not a marker for proximal tubules but loops of Henle/distal nephrons.

4. In figure 5B, the similarity between organoids and human samples is explained. However, the commonly expressed genes are only 10, and the majority of TNF- signature genes are not expressed in organoids. The differences should be also explained.

5. The VCAM1 expression in figure 6C in TNF α - treated organoids is seen in stroma, podocytes, and tubules. However, figure 6E of human kidneys show the VCAM1 expression in DTL. This result appears to highlight the limitation of the organoid model used in this study. The mechanism of TNF α -induced VCAM1 is also unclear.

6. VCAM1 is expressed in normal Bowman's capsules (PEC) in humans. Figure 6E shows the VCAM1 expression in DTL but not in PEC, and it is also unclear if TNF α induced the VCAM1 expression in DTL.

7. While the NEPTUNE data is from FSGS/MCD patients, the current manuscript does not show podocyte phenotypes in organoids treated with TNF α . Hence, it is unclear if the organoid model recapitulates the patient pathology of FSGS/MCD.

Minor:

1. In lines 144-163, please write the time points of organoids assessed in this section.
2. In lines 165-167, the cited paper (ref # 24) is not a peer-reviewed paper.
3. In line 183, figure 4E?
4. In line 199, Fig. 4G?

Reviewer #3 (Remarks to the Author):

Lassé et al have conducted a study that combines proteomics and transcriptomics (in bulk and single cell) to study the gene expression changes of kidney organoids during extended culture and disease modeling. Through differential gene analysis, they showed, in both mRNA and protein levels, that extended culture leads to loss of kidney cell signature and accumulation of extracellular matrix in kidney organoids. By comparing the protein profiles between organoids and the adult kidney or immortalized cell line, the authors demonstrate that the kidney organoid is a better cell source than the immortalized cell line for use in kidney disease modeling since it captures more kidney cell type signatures than the immortalized cell line does. Then the authors turn to use scRNA-seq and bulk proteomics to assess the disease modeling capacity of kidney organoids with TNF- α treatment. The authors identified a set of TNF- α responsive genes from kidney organoid and demonstrate that this gene-set can be used to as biomarkers to classify the human samples from kidney diseases.

Overall, the proteomics datasets present by the authors in this manuscript can serve as good resources to the scientific community since not many protein profiling datasets are available in the public database. The idea to use proteomics combined with bulk RNA-seq and scRNA-seq to monitor the organoid differentiation and assess disease modeling is quite novel and exceptional. However, I have several concerns about data interpretation and analysis as specified below. Hopefully these comments can help the authors improve their manuscript in the current form.

1) Kidney organoids hold great promise for disease modeling but its use for that purpose also suffers several drawbacks. For example, the key differences between kidney organoid and adult human kidney are that kidney organoid cell types are not mature and that kidney

organoid contains non-kidney cell types such as neurons and muscle cells. When the authors interpret the data from the analysis, they should keep in mind that these differences can perplex the data interpretation especially when the proteomics dataset is from bulk profiling (not single cell) where the gene expression represents an average expression of all cell types including the immature cells and off-target cells.

I suggest the authors to perform additional analyses to help clarify this concern. For example, when the authors compare the proteomics between kidney organoid and adult kidney cells (microdissected tubule and glomeruli), do they see the more developmental genes or neuronal genes expressed in the kidney organoid? Are the off-target cells such as neurons and muscle cells also respond to TNF- α treatment (based on the scRNAseq data).

2) There are no human kidney diseases solely caused by TNF- α . That raises a question as to whether the use of TNF- α treatment in organoid is a good approach to model kidney diseases. To establish the link between TNF- α and kidney disease, the authors should at least show that TNF α signaling (TNF- α and its downstream targets) is upregulated in most forms of the kidney diseases.

3) The most intriguing point from proteomics data is that the correlation between protein and RNA expression is only less than 0.25, which means many genes are discordant in their RNA and protein levels. The authors should show a heatmap to highlight the differential genes that are only detected in protein but not in mRNA.

4) In human kidney diseases, Vcam1 and C3 are upregulated in the PT and immune cells not in stroma. In Figure 6C, both Vcam1 and C3 are upregulated in stroma cells from kidney organoid. Would that indicate the difference between organoid and adult kidney in terms of cell type responses?

5) Figure 6E shows that DTL responds to TNF- α treatment strongly. Since no much are known for the role of DTL in kidney disease, the authors should do additional work to validate this result. E.g. immunostaining of TNF receptor and the DTL marker to validate that this cell type can respond to TNF- α signals.

6) In Figure 5e, there are 320 genes in total but when breaking down by secreted and non-secreted proteins, the authors reported $302 + 22 = 324$ genes. What are the additional 4 genes from?

7) In Figure 5g, the authors should repeat the analysis by using same number of cells to rule out that the cell response was skewed due to unequal n of cells contributed by each group. I raised this concern because I noticed that the most responsive cell types are the cell types have greater difference in cell number between groups (for example, 2600 vs 30 cells for the immune cell type).

8) The authors need to explain the jargons used in the manuscript. For example, what are 2D keyword enrichment and the combined score in the pathway analysis?

RESPONSE TO REVIEWER COMMENTS

We thank the reviewers and editors for your thoughtful reading and constructive critique of our manuscript. Please see our responses in bold.

REVIEWER 1

Lasse M et al

Nature Communications 22-38308

An integrated organoid omics map extends modeling potential of kidney diseases

Summary

Kidney organoids, especially those derived from patient specific pluripotent stem cells, are thought to be an improved model to study kidney disease over the classical monoculture or co-culture systems.

Lasse M et al present an in-depth interrogation using primary bulk proteomic, bulk RNAseq, and sc-/sn-RNAseq transcriptomic data sets derived from organoids cultured for specific time points as well as organoids stimulated by cytokines known to be associated with progressive kidney diseases. In addition to primary data the authors integrate scRNAseq and snRNAseq datasets available through public consortia (ECRB and Neptune). The organoid model recapitulates finds expected for expression of ECM with aging and increase relative activity of stromal cells over podocyte-like cells with aging. Stimulation of organoids with TNFalpha increased expression of 322 gene products many that are also associated with poor outcomes in patients with proteinuric diseases/syndromes. Two proteins, complement C3 (C3) and Vascular cell adhesion protein 1 (VCAM-1) previously identified as proteinuric disease theranostic biomarkers highlighted the potential for kidney organoids use in biomarker engagement studies.

Comments & Critiques

Lasse M et al present a tour de force of studies addressing the functional relevancy of kidney organoids to the study of human proteinuric diseases. Key points in the manuscript are (1) defining the expressional programs in organoids with transitioning across the glomerular phenotype into an aged phenotype, (2) integrating the proteomic and transcriptomic datasets, (3) comparing the organoid expressional programs with RNAseq data sets for microdissected microstructures (glomerular, tubular) from patient biopsies, (4) defining organoid response to cytokine (TNFalpha) stimulation, (5) integrating organoid TNFalpha response with imputed TNFalpha response results from patient data, and (6) a targeted analysis of TNFalpha responsive proteins, C3 and VCAM1, to define patient response.

(1) Analysis of kidney organoid trajectory

Summary: Previous studies of kidney organoid development documents a shift in protein expression toward glomerular differentiation at 3wk. Therefore, the shift in proteome and transcriptome was followed at 21d and 29d. Large numbers of proteins were identified (6,700) with sufficient information for quantitative analysis on 5,400 proteins. Differential protein abundance between 21d and 29d was observed for 350 increasing and 428 decreasing proteins. The changes for marker proteins were used as surrogates to infer cell specific or compartment specific changes such as SYNPO or NPHS1 for podocytes and ACTA2 and PDGFRA for extracellular matrix compartments.

Critique: Why are ACTA2 and PDGFRA used as markers for ECM? Figure 1D documents many GO processes specific to collagens, laminins, and other proteins that would be a better phenotype for fibrotic diseases? If this has not been defined then it should be added at some contextual level. If the PDGFRA represents a unique aspect of profibrotic disease and ECM upregulated that should be acknowledged.

We thank the reviewer for the opportunity to clarify. Our original intent for the immunostainings shown in Fig. 1B was to provide validation of proteomics data. We chose PDGFRA and ACTA2 based on their prominence as differentially expressed proteins in our data and as markers of profibrotic state; both are thought to contribute to

fibrogenesis associated with kidney disease¹ and to be expressed by organoid stromal cells/myofibroblasts following treatment with TBFb1².

However, we recognize that extracellular matrix components COL1A1 and FN1³ have been shown to increase in organoid models of IL-1b-induced fibrosis⁴ and TGFb1-induced fibrosis⁵. Accordingly, we performed additional IF stainings for COL1A1 and fibronectin (FN1). Both proteins also increase in the stromal space with increased culture duration (D29 vs D21 as shown below in *Reviewer Fig. 1* (now included in *Suppl. Fig. 1B/C*). Based on our combined proteomics and transcriptomics analysis, organoid stromal cells appeared to express the majority of COL1A1 and fibronectin (FN1) (See *Fig. 2E*).

Reviewer Fig. 1: Additional stainings for kidney organoid extracellular matrix proteins.

(2) Organoid proteome-transcriptome integration and cellular origins of proteins

Summary: Bulk transcriptomic expression changes were used to assess regulation of gene expression and then correlated to protein abundance. Unsurprisingly the bulk correlation was low but consistent as measured by a word enrichment. scRNAseq analysis of organoids resulted in 14 cell specific clusters with a majority as kidney specific. Annotation of the 14 cell specific subtypes on to the proteomic data noted the largest proteome changes occurred in the stromal cell population and decreases in cells corresponding to early glomerular epithelial cells and podocytes. To better address correlation of gene and protein signatures the authors took the top 30 differentially abundant gene products and integrated expressional changes. The majority (90%) were renal cell specific observations decreased over time in relative abundance at both the transcript and protein level. The majority of the stromal cell specific observations increased over time.

Critique: What is word enrichment? Is this a feature for analysis of GO annotation data? At this time it would appear to be a word cloud analysis? Is this the most robust analysis of GO terms?

We thank the reviewer for the opportunity to clarify our terminology and choice of analytical tools. We would like to assure the reviewer that this is not a word-cloud analysis. Rather, “Keyword” refers to uniprot keywords (<https://www.uniprot.org/help/keywords>), an alternative to Gene Ontology (GO) annotation for categorizing proteins. More details have been added to the description in the Methods section. 2D GO term analysis can be used to investigate multi-omic data⁶. We have clarified this in the figure legends and methods.

The authors data suggest the largest (top30 changes up or down in gene/protein expression) are not regulated post-transcriptionally. Is there a common feature of upstream gene activators such as TF that might inform of gene expression activation? How are these TF integrated into TNFalpha signaling?

Regarding top30 gene/protein changes, we presume that the reviewer is referring to the data in Fig. 2E. Our intent in this figure was to highlight the largely consistent expression patterns between transcripts and proteins for the top 30 genes. However, we agree with the reviewer that these RNA transcripts show co-directionality on the protein level (i.e., post-transcriptional). So, we explored this further by generating comparison plots as shown below in Reviewer Fig. 2 (now included in Suppl. Fig. 2A) with examples of wide variance in protein expression that was not appreciated in transcript expression. We have modified the text accordingly.

Reviewer Fig. 2: Organoid genes with dissociation of protein and transcript abundance.

To address gene expression patterns observed in maturing organoids, we identified upstream regulators of genes differentially expressed between days 29 and 21 in our two bulk RNA-seq organoid time course datasets (Suppl. Tables 3,14) using Ingenuity Pathway Analysis. 107 upstream regulators were identified as common to both datasets shown in Reviewer Fig. 3 below (now included in Suppl. Fig. 3F and Suppl. Table 16), 30 of which were transcriptional regulators. Given the complexity of transcriptional regulation, significant future effort will be required to investigate control of the maturing organoid gene expression program. Consistent with the pro-fibrotic protein expression pattern in maturing organoids (Fig. 1), the list of 107 upstream regulators also contains three TGF-beta components (TGFB1, TGFB1R and TGF beta) with highly enriched TGFB1 mechanistic networks connected to 174 genes and 228 genes in datasets #1 and #2 accordingly (now included in Suppl. Table 17).

Reviewer Fig. 3: Upstream regulator analysis of maturing organoids identifies 30 upstream transcriptional regulators and highly enriched TGFB1 mechanistic networks.

We then explored the overlap of upstream regulators in the above analysis with DEGs identified in our TNF vs. control dataset (now included in Suppl. Table 15). 41 shared upstream regulators were identified, 11 of which were transcription factors shown in Reviewer Fig. 4 below (now included in Suppl. Table 18, Suppl. Fig. 3G). Classical integrators of inflammation, such as NF kappa B, were activated in maturing as well as TNFalpha-treated organoids; in TNFalpha-treated organoids, this effect was much stronger. To assess the involvement of the canonical TNFalpha signaling pathway observed in the TNFalpha-treated organoid proteome, we also mapped differentially expressed proteins in this pathway (now included in Suppl. Fig. 4E) as discussed in the revised text.

Reviewer Figure 4: Comparison of upstream regulators in maturing and TNFalpha-treated organoids.

(3) Comparison of organoid proteome organization with mature human tissue

Summary: To better understand the expressional programs of the organoids the authors integrated the proteomic data of the organoid with the proteomes of isolated human glomeruli, isolated human tubules, and immortalized podocytes. GO terminology for the comparisons organoid to glomerular or tubular microstructures were enriched in terms associated with receptor mediated endocytosis and ECM. Podocytes were enriched uniquely in terms associated with metabolism.

Comment. Is the integration of the podocyte data here relevant as the GO terms enriched would suggest single cells in culture are ontologically enriched in terms for metabolism and not necessarily for terms associated with higher order interactions with the 3D environment. The authors note that organoids are superior to podocytes in culture perhaps that is too passive of a statement.

We agree with the reviewer; the higher order interactions present in multicellular 3D kidney organoid cultures generate podocytes that are clearly superior to those generated in 2D monocultures with respect to their gene expression. However, stem cell derived organoids are both time and resource intensive and may not be required, nor sufficient, for specific experiments. In these cases, pure cell type population or monolayer can be used to study protein biochemistry or function, and manipulation may be more straightforward. With these factors in mind, we strove to compare expression data between podocyte monocultures and multi-cell type 3D organoids to provide a resource for researchers to use in future experimental design. Since Reviewer 3 expressed interest in this representation, we opted to keep these data in Fig. 3C/D, but we have revised Suppl. Fig. 3C-H and updated the relevant text for improved clarity.

To further explore the intercellular interactions occurring in the 3D organoid environment, we used NicheNet⁷ to generate an in silico-interactome map based on expression of expected downstream effectors following ligand-target engagement, based on the significant differentially expressed protein expression list from Suppl. Table 2 assigned to cell type (Fig. 2D). These data shown below in Reviewer Fig. 5 (now included in Suppl. Fig. 2B and relevant text).

b

Potential receptors expressed by cell type

Reviewer Fig. 5: NicheNet analysis revealing functional ligand-target engagement.

We also used CellChat⁸ to investigate potential ligand-receptors pairs using the list of all proteins detected in organoids (Suppl. Table 1) and assigned to cell types as in Fig. 2D. As shown, the results suggest a rich environment of potential cell-cell interactions. However, as this analysis is limited to a description of ligands and targets expressed by organoid cells without the functional evidence of target engagement achieved with the NicheNet analysis above, we chose not to include it in the manuscript.

Reviewer Fig. 6: CellChat analysis for potential ligand-receptor pairs.

(4) Organoid responds to TNFalpha with activation of proinflammatory proteins

Summary: Prior investigations documented the response of kidney organoids to TNFalpha and the correlation of differential gene expression with clinical populations for FSGS and MCD. Key targets observed in this study were CCL2/MCP-1 and TIMP1. The kidney organoids expression of TNFalpha receptors was observed only at the gene level and only significantly for TNFRSF1A. Immunofluorescent studies suggested TNFRSF1A expression would be specific to N-cadherin expressing (proximal tubular) cells. The organoids responded to TNFalpha to alter the expression of approximately 300 proteins including increased expression of 145 proteins and decreased expression of 157 proteins. Included in the increased are VCAM-1, NF-kB2, and ITGA3; all markers of early glomerular epithelial cells.

Critique: Please address why expression of TNFRSF1A in proximal tubular cells impacted expression of glomerular epithelial cells.

It was not our intent to suggest that expression of TNFRSF1A (TNFR1) in PT cells impacts expression of GEC proteins. Rather, our original intent was to demonstrate by IF that kidney organoid cells expressed TNFR1 despite protein not

being detected by proteomics (Fig. 4C). To clarify this, we originally included scRNA-seq data demonstrating that other organoid cell types expressed TNFRSF1A transcript including maturing podocytes, and two clusters of early glomerular epithelial cells (Fig. 4B). As shown below in Reviewer Fig. 7, we have now added additional IF images of our organoids demonstrating TNFR1 expression in podocytes (SYNPO+) (now included in Fig. 4C). These data suggest that podocytes as well as other organoid cell types are capable of responding directly to TNF α stimulation, leading to increased expression of NF- κ B2, ITGA3 and VCAM1. This concept is further supported by organoid single cell transcriptional data shown, plus IF (see Reviewer Fig. 12 below) (now included in Suppl. Figs. 4A-C).

Reviewer Fig. 7: Expression of TNFRSF1A in podocytes, stromal and tubular cells.

Consistent with prior knowledge that TNFR1 canonically signals through NFKB^{9,10}, we see evidence of activation of NFKB target genes and protein expression in organoids in response to TNF. Moreover, NF kappa B is prominently included in new our upstream regulator analysis (now included in Suppl. Table 18, Suppl. Fig. 4G). Thus, our data suggest that TNFR1 signaling likely occurs in multiple organoid cell types via established TNF signaling mechanisms. Relevant text was updated.

Also please justify the use of the secretome data. A large amount of protein was analysis using state of the art methods and the secretome analysis yields a paucity of protein identification (120 on average). Moreover, a large number of these identification are naturally low abundant and yet readily observed. How was the presence in the growth media or as ECM protein components intrinsic to the organoid culture system eliminated? The authors have to address how these data are robust and specific to the organoid and not related to experimental conditions or treatments.

The paucity of proteins identified in the organoid secretome can be explained in large part through the relatively high concentrations of proteins which are inherent to the medium required for extended organoid culture, such as human serum albumin and transferrin (Gene = ALB, TF). These high abundance proteins in media can limit the detection of low abundant signals. Since these components are required for long-term culture viability, we evaluated samples in the usual media.

Initially, samples of medium containing protein supplements (not exposed to organoids) served as controls. To address the concern of the reviewer that components of the organoid culturing system may influence results of differential secreted protein expression, we have now included additional controls.

These controls are:

1. Proteomics analysis of the culture media from Geltrex-coated wells, treated with all components according to normal culture protocol in parallel with other wells, but absent of cells (acellular supernatant)
2. Proteomics analysis of the pure Geltrex matrix mixed with DMEM media (as used to coat wells for organoid culture). Please note that this is the matrix in extremely high concentrations.

The acellular culture media contained 22 proteins, similar to our previous control experiment. The Geltrex matrix with DMEM contained 54 proteins. We then analyzed the overlap between these controls and our previously measured differentially expressed proteins (TNF medium, see upset plot). Of the 22 regulated proteins upon TNF α stimulation, no proteins were detected in the acellular supernatant while 4 proteins were also detected in Geltrex-DMEM (FN1, HSPG2, LAMB1, SPARC).

However, contribution of these 4 proteins (HSPG2, FN1, LAMB1, SPARC) to the outcome of our differential expression analysis is mitigated by our experimental design: comparison was made between TNF-treated and vehicle control-treated organoids at the same time-point under the same conditions. Additionally, and not surprisingly, we detected robust expression of transcript corresponding to these 4 proteins indicating that organoids also express them. Thus, we argue that our appropriately controlled secretome data is relevant. These data are now included in *Suppl. Fig. 3A/B*.

(5) Proteome alterations in TNF α -induced organoids help stratify disease human kidney tissue

Summary: The authors sought to translate the organoid findings into human disease by comparing the signature of gene activation to previously characterized TNF α -associated changes in 272 genes associated with poorer outcomes in MCD and FSGS populations. The authors focused on organoid renal cell populations and sought to exclude stromal cell populations. The gene signature was expanded from 272 to 322 gene products. Within those two groups the authors focused on 10 gene products for elaboration and discussion (five of the 10 proteins were resident in the TNF α responsive secretome data). GO analysis of the 272 gene products suggests comparatively to the 322 gene products a focus on metabolism rather than ECM and extracellular cytokine signaling. Application of the 322 gene signature to the kidney tissue transcriptome cluster identified an association with a patient cluster having poor outcomes to therapy. The weak association of the organoid signature with the patient data suggests patient data is more granular and complex. Thus perhaps the organoid data can drive subgroup analysis.

Critique: Correlation of TNF α responsive expressional programs is important for understanding outcomes in the targeted patient populations. As of now it is unclear how this process was performed. The changing between 272, 322, and 10 gene products is confusing. It appears that at some states these analyses were independent, as the 272 were completely part of the 322 or that there are only 10 gene products overlapping. Do these two panels remain independent during the analyses or were the 322 genes all-inclusive of the 272 plus some 48 (not 10) additional gene products? What is the enrichment of pathways in the entire 48 overlapping genes?

We thank the reviewer for pointing to our need to clarify our approach. We compare two separately derived TNF signatures and have updated *Fig. 5A* for reference, see *Reviewer Fig. 8* below:

- 1. A kidney tissue 272-gene signature (Tissue TNF signature): based on computational predictions from differential gene expression in transcriptional profiles of kidney tissue from individuals in the NEPTUNE cohort of FSGS/MCD, as previously published^{11,12}.**
- 2. An organoid 322-gene signature (Organoid TNF signature): based on experimentally measured protein expression, i.e., proteomics analysis of organoids following TNF α stimulation. (2 peptides originally were not mapped, now updated and added to the gene signature in this revision).**

a TNF Signatures

Organoid (322 genes)

Experimentally-derived from differentially expressed proteins in TNF-treated kidney organoids (Fig. 4D/F)

Tissue (272 genes)

Computationally-derived from transcriptomes of 220 NEPTUNE kidney biopsies (adapted from Mariani et al., 2023)

Reviewer Fig. 8: Clarification of TNF Signatures.

Ten (10) genes are included in both signatures; we have relabeled the 10 genes as “overlap” and clarified this in (revised position) Fig. 5C (Reviewer Fig. 9 shown below) as well as in the text. We explored this seemingly limited overlap of genes in the original submission by asking whether additional genes from both signatures could be contributing to networks common to both signatures. To answer this question, we performed pathway analysis on the combined the 2 signatures (total 584 unique genes = 272+322-10 overlap). As shown in Fig. 5D (moved position from original manuscript), additional genes from both signatures contribute to a TNF-NFkB network, demonstrating a functional overlap of the signatures that is not captured solely by direct comparison of lists of genes. Relevant text was updated.

Reviewer Fig. 9: Clarification of 10 overlap genes in TNF signatures.

(6) TNFalpha-dependent molecules C3 and VCAM1 can stratify disease kidney tissue
 Summary: The kidney organoid and patient RNAseq data sets overlapped in the TNFalpha gene response signature with 10 gene transcripts. Two transcripts of prior interest due to associations with pro-inflammatory character were complement C3 and VCAM1. Both C3 and VCAM1 transcripts and protein increased with TNFalpha stimulation, and C3 abundance in organoid cell culture media were increased. Examination of C3 and VCAM1 expression in snRNAseq data from human biopsy material localized C3 and VCAM1 expression to the distal thin limb tubular cell types and treatment of micro-dissected DTL tubular cells with TNFalpha recapitulated the organoid expression changes.

Critique: These data are intriguing as C3 and VCAM1 are relevant and known markers for renal diseases. What is a surprise is the relationship to the DTL.

Thank you for highlighting this key concept. To validate these findings, we have now performed staining in human kidney tissue for VCAM1 in descending thin limb (DTL). First, we note that our antibody recognized VCAM1 expression in the parietal epithelial cells as well as the proximal tubule (LTL+/AQP1+) as expected as shown below (now included in *Suppl. Fig. 8B*).

Reviewer Fig. 10: VCAM1 expression in human kidney tissue, PEC (left panels) and PT (right panels).

Next, we examined tissue from 9 individuals with FSGS diagnosis. The quality controls for these stainings are now found in *Suppl. Fig. 8A*, demonstrating that AQP1positive/LTL negative cells in the outer medulla are DTL cells. As shown below (now included in *Fig. 6F* and *Suppl. Fig. 8C*), individuals with low eGFR and high proteinuria (pts #2,4) demonstrated clear VCAM1 expression in DTL segments (AQP1+/LTL1-). VCAM1 positive DTLs cannot be seen in corresponding segments in tissue from individuals with high GFR (pts#1,3). Given that C3 is also a circulating liver protein, we limited our human tissue analysis to VCAM1 expression. The relevant text for these findings was updated.

Reviewer Fig. 11: VCAM1 expression in DTL in low GFR FSGS (pts #2,#4) but not high GFR FSGS (pts #1, #3).

Overall critique: The authors have done a masterful job addressing the value of kidney organoids as a surrogate for the study of human kidney disease. Data are presented for (A) differential expression of proteins between the 21d and the 29d program to be stromal cells and glomerular cells resulting in the enrichment in stromal cells and glomerular cells as sources of protein expression; (B) regulation of gene expression programs by TNFalpha the authors note the expression of TNFRSFA1 by proximal tubular cells, and (C) TNF-gene response program marker components C3 and VCAM1 to map expression to the DTL.

(1)How did the age of the organoid at treatment time for (B) and (C) compare with (A). (2) How do the authors explain the changes to the TNFalpha responsive program taking place in the tubular compartments and not the glomerular

compartment or the stromal compartments? Is it possible to experimentally spatially resolve organoid DTL, PTEC and glomerular expression for TNFRSFA1 (FISH) with C3 OR VCAM1 with SYNPO or NPHS1 with ITGA3?

Thank you for your positive critique. To address your questions:

- 1) The age of TNFalpha treated organoids was D24/D25, so falls in the middle of the studied time course (days 21-25-27-29). We chose this time point because of the clear expression of TNFRSFA1 at D24-25 by scRNA-seq and immunostaining, but also before any potential overwhelming ECM deposition or de-differentiation. This is now highlighted more prominently in the figure legends and text.
- 2) As demonstrated in the immunostaining of organoids in *Reviewer Fig. 7* above, TNFRSFA1 expression is widespread in organoid cell types and includes podocytes (SYNPO), stromal (MEIS1/2) and tubular (CDH2) cells, which is consistent with our scRNA-seq data. Regarding spatial resolution of TNFalpha-response genes, we see ITGA3 and C3 in podocytes (SYNPO) as well as VCAM1 in podocytes (PODXL) as shown below in *Reviewer Fig. 12* (now included in *Suppl. Fig. 4A/B*). Relevant text was updated. Additionally, based on cell morphology, all 3 are expressed in PECs, while C3 and ITGA3 are expressed in tubular cells. VCAM1 expression was also detected in TNFRSF1A-expressing cells (panel B). We were unable to discern the DTL cell type (AQP1+/LTL-) in organoids, presumably due to the relative immaturity of the organoids (see also response to Reviewer 2). To disentangle the exact direction and 3D-control of the transcriptional programs will be an area of future investigation.

Reviewer Fig. 12: Immunostainings of C3, VCAM1 and ITGA3 expression in in TNF-alpha treated organoid podocytes.

REVIEWER 2

This study is to characterize kidney organoids by proteomics and transcriptomics and validate the organoid model to study TNFalpha-mediated renal disease with data from NEPTUNE study. Overall, **the manuscript contains nice data sets that would be helpful for the research communities**. However, **lack of interventional/mechanistic studies on TNFalpha-induced podocyte diseases weakened the scientific rigor** and impact of the study. The followings are my suggestions for improvement.

We thank you for your constructive critique. We agree that mechanistic studies would further the goal towards understanding kidney disease pathomechanisms. However, much is still to be determined regarding the ground truth

of organoids, e.g. significant scRNA-seq data are now available, but minimal proteomics data are publicly available. To build on our original goal to create a resource and assessment of the translational value of the organoid model, we have added the following experimental results in this revision:

- Additional mechanistic studies, using a small molecule TNFRS1A signaling inhibitor, demonstrating the initial signaling mechanism for the presented key findings
- Additional focus on rigor and replication, by confirming key TNFalpha response generated from:
 - 1 additional human male iPSC line, using the same organoid protocol
 - 1 additional human female iPSC line, using a different organoid protocol

Major:

1. **Batch-to-batch variations are one major challenge** in organoid research. **Only one line of human pluripotent stem cells is used** in this study, and it is **unclear how many independent experiments** were performed.

1. **Batch-to-batch variation:** We agree with the reviewer regarding batch-to-batch variability of organoid experiments, and we exerted significant effort to address this issue. We initially included crucial information regarding experimental numbers in the figure legends and Methods, but have clarified in the revised text, Methods and figure legends. Note that some samples were limited (e.g. organoid time course RNA-seq) by the material available when performing parallel assessments (i.e. proteomics and bulk RNA-seq). Overall, we found highly consistent behaviours across batches.
 - a. For the organoid time course and TNFalpha proteomics, we generated data using biological (not technical) triplicates. 2 independent experiments were performed for each, with consistent findings. This is a common number in proteomics studies where large differences are expected.
 - b. For the TNFalpha-treated organoids, we used data generated from 5 independent experiments (*Fig. 4G*, data from only 3 organoid lysates were shown, as the samples from the other 2 experiments were used for proteomics).
 - c. For the organoid time course bulk RNA-seq analysis, data from 2 independent experiments are included (*Suppl. Tables 3 and 14*). For the TNFalpha bulk RNA-seq, 1 experimental set was sequenced (*Suppl. Table 15*) after confirming inter-experimental consistency by qRT-PCR and ELISA (*Fig. 4G*).
 - d. For the new male iPSC line organoid experiments, we performed 3 independent experiments (batches).
 - e. For the new TNF pathway inhibitor experiments, we performed 3 independent experiments (batches).
2. **Validation:** As recommended by the reviewer, we have added new data (shown below) in which we validated key findings from our original experiments by generating organoids from 2 additional iPSC lines, 1 using a different differentiation protocol. Ideally, we would have used the original line with the new differentiation protocol, but the allotted time frame for revision did not allow us to establish necessary institutional regulatory approval for sharing human PSC lines in time to perform the necessary experiments.
 - a. Organoids generated from a second iPSC line from a NEPTUNE (male) participant using the original differentiation protocol. We found that they respond very similarly with regard to VCAM1, CXCL10, and C3 gene expression and secretion in response to TNF alpha (now found in *Suppl. Fig. 7A/B*).
 - b. Organoids generated from a third human iPSC line (female) using a second differentiation protocol (see next reviewer comment for further details). We found that they respond similar with regard to VCAM1, CXCL10, and C3 secretion in response to TNF alpha (now found in *Suppl. Fig. 7C/D*).

Reviewer Fig. 13: Validation of key TNF α -induced proteins in organoids generated from additional human iPSC lines (panel B on left, panel D on right) and a different differentiation protocol (panel D).

2. There are several differentiation protocols for kidney organoids, yet only one protocol is used in this study. The rationale for the selection of the differentiation protocol is unclear. Differentiation protocols that do not follow the metanephric kidney development resulted in poor differentiation and fibrosis development in short-term culture (e.g. PMID: 30033089). Indeed, fibrosis is induced in 4 weeks of differentiation in this study as shown in figure 1.

We thank the reviewer for raising this issue. We used this differentiation protocol because our previous in-depth characterization demonstrated similarities to developing human kidney cells¹³. In that work, we demonstrated alignment of the kidney cell-type specific developmental trajectories in our organoids with the corresponding cell types in developing human kidneys, including podocytes, stromal and tubular cell types. This is a well-recognized organoid protocol^{14,15}. Comparison to other well-recognized organoid protocols¹⁶ demonstrates that our protocol generates a similar distribution of nephron-related cell types including podocytes. We chose a time point with relatively little fibrosis (D24-25) for the TNF α experiments, based on our time course data. The relevant text was updated.

Further, we performed additional experiments using organoids generated using a second protocol from this comparison group, with well-characterized metanephric kidney development^{17,18}. Targeted protein expression analysis confirmed that these organoids responded to TNF α in a similar manner, as shown above in item #1. Though performing a complete organoid omics map was beyond the scope of this revision, initial targeted proteomics analysis (shown below in Reviewer Fig. 14) revealed a comparable proteome, and concordant regulation of the 10 overlap proteins (Fig. 5C). However, we agree with the reviewer that different organoid differentiation protocols could markedly impact findings and have now emphasized this point as a limitation in the Discussion.

Reviewer Fig. 14: Heatmap demonstrating similar expression of panel of 10 overlap proteins in organoid proteomes generated using different differentiation protocols.

3. In the line174, I believe authors meant E-cadherin (CDH1). CDH1 is not a marker for proximal tubules but loops of Henle/distal nephrons.

Indeed, we meant N-cadherin (CDH2), not E-cadherin (CDH1) and thank the reviewer for catching our error. This has been corrected in Fig. 4C as well as in the corresponding text.

4. In figure 5B, the similarity between organoids and human samples is explained. However, the commonly expressed genes are only 10, and the majority of TNF- signature genes are not expressed in organoids. The differences should be also explained.

We thank the reviewer for the opportunity to clarify this critical concept, which was also recognized by Reviewers 1 and 3. We identify several factors which contribute to the observed difference between the 272-gene Kidney Tissue and 322-gene Organoid TNF signatures (see updated Figs. 5A/C, Reviewer Figs. 8-9 above):

1) We are comparing gene signatures derived from transcriptional and proteomic datasets: We note that almost 74% of the 272 genes in the Tissue TNF signature are transcriptionally expressed in TNF-stimulated kidney organoids. However, the actual number of total proteins detected (almost 7000) is lower than the transcripts captured by RNA-seq (above 20,000). This suggests that there is likely a number of relevant proteins which we are not detecting. Since transcript level does not necessarily inform protein expression (confirmed by us in Fig. 2B; also see response to Reviewer 1, item #2), this is likely contributing to a difference between the signatures. Indeed, we detect about 50% of the Tissue TNF signature genes on a protein level but less than 10% are differentially expressed.

2) The signatures are based on differential gene expression: Genes that were expressed, but not significantly differentially so between conditions, were not included. As rigid differential expression cut-offs were used in the generation of both signatures such that signature inclusion was limited to 1.5-4.5% of detected gene products, we expect this also impacts gene expression overlap.

3) Organoids contain limited cell types: A lack of vasculature, distal structures, and immune cells plus the relative immaturity of the cell types greatly impacts the cellular environment and resulting omics signatures. In comparison, the signature derived from human tissue is sampling expression in dozens of cell types, kidney and immune. Thus, organoids are focused on the cellular events of a subset of human kidney cells.

4) Pathway analysis confirms shared contribution of genes from both signatures to key networks despite a limited overlap of individual genes: Fig. 5D highlights the contribution of additional genes (in addition to the 10-gene overlap in Fig. 5C) from both signatures to the TNF-related network.

We chose to explore kidney organoid protein expression to ensure capture of a robust TNF response, though protein capture is prone to limitations in coverage as compared to transcript-based methods. It is poignant that our protein-based Organoid TNF score was able to capture gene activity not captured by the Tissue TNF signature, but clearly relevant to FSGS/MCD (Fig. 5B/D-H and Suppl. Fig. 5B). The differences in the gene networks captured by each gene signature clearly reflect these two different strategies (Suppl. Fig. 5A). Focused attention on expression of the 10 overlap gene set, however, presents an opportunity to prioritize genes within the Tissue TNF signature for exploration as biomarkers used to align clinical response with personalized modeling of kidney disease *ex vivo*.

5. The VCAM1 expression in figure 6C in TNFalpha- treated organoids is seen in stroma, podocytes, and tubules. However, figure 6E of human kidneys show the VCAM1 expression in DTL. This result appears to highlight the limitation of the organoid model used in this study. The mechanism of TNFalpha-induced VCAM1 is also unclear.

1. VCAM1 expression in kidney cell types:

We appreciate the reviewer highlighting the discrepancy between organoid and human tissue VCAM1 expression; this point was also raised by Reviewer 3, item #4. In *Suppl. Fig. 5C/D*, we see widespread but variable VCAM1 expression in organoid cell types in response to TNFalpha, and a similar pattern in high TNF status human kidney tissue-derived cells. Based on snRNA-seq (*Fig. 5G*), VCAM1 is expressed in many cell types in the human kidney samples but the most significant differential VCAM1 expression appears to be in DTL. We have now clarified that VCAM1 protein is expressed in the DTL cells (AQP1+/LTL-) of FSGS-affected kidney tissue on protein level as described above and added this to the manuscript (*Fig. 6F, Suppl. Fig. 8C*; see also *Reviewer Fig. 11* above in Reviewer 1 item #6).

The specific location of differential expression seen in the adult kidney is not clearly replicated in the organoid model due in part to the relative immaturity of the cell types (we were not able to identify LTL-/AQP+ tubular segments). However, organoids still reveal a robust and reproducible VCAM1 expression with TNF stimulation. So while we agree that organoid models are limited by immature cell types, our organoid model still appears to have identified an important disease-relevant marker. Further, given the potential for kidney organoids to augment human disease research, we argue that defining these limitations can aid efforts to improve existing protocols. This thought is now added in the discussion.

2. Mechanism of TNFalpha-induced VCAM1 expression:

We performed additional experiments and analyses in organoids to address the mechanism of VCAM1 and TNFalpha signaling.

First, we clarified whether TNFR1 signaling (via TRADD/RIP1) is the key mediator of TNFalpha-stimulated increase in VCAM1 expression. Increase of VCAM1 could be secondary to activation of CXCL10, C3 or other off-target effects in the supernatant. To confirm that this occurs through TNFR1 (TNFRSF1A) signaling, we inhibited the TNF pathway using a commercially available small molecule inhibitor R-7050 that prevents TNF receptor-adaptor molecule complex formation (TNFRSF1A – TRADD – RIP1) and subsequent receptor internalization (<https://www.sigmaaldrich.com/US/en/product/mm/654257>)¹⁹. Co-treatment of inhibitor with TNFalpha stimulation revealed robust suppression of VCAM1 and C3 expression. These results are shown in *Reviewer Fig. 15* below (now included in *Fig. 6C*, relevant text updated).

Reviewer Fig. 15: TNF pathway inhibition blocked C3 and VCAM1 expression in TNFalpha-treated organoids.

We attempted to explore this pathway further using NFkappaB inhibitors, however these were toxic to cell cultures. This could be related to the developmental immaturity of the organoid cells & NFkappaB's importance during development²⁰. Thus, further experimental iterations will be necessary to investigate signaling and toxicity of NFkappaB inhibitors in the context of human organoids.

We also analyzed the mechanistic pathway. Our organoid proteomics analysis revealed very high abundance expression of VCAM1, in addition to NFKB2 and other upstream regulators following TNFalpha treatment (Suppl. Tables 5,7). We mapped these data onto the canonical TNF signaling pathway as shown in Reviewer Fig. 16 (now included in Suppl. Fig. 4E). Upstream regulator analysis also confirmed the prominent involvement of NfkappaB following activation by TNFalpha in two different organoid datasets as seen in Reviewer Fig. 4 in Reviewer 1 item #2 (now included in Suppl. Fig. 4G, Suppl. Table 18). Relevant text was updated.

Reviewer Fig. 16. TNFalpha-stimulated expression of proteins in the canonical TNF signaling pathway in organoids.

Together, our findings are consistent with long-standing understanding of the canonical signaling of TNFalpha-induced VCAM1 expression, with NfkappaB as a central integrator to control surface expression of VCAM1²¹⁻²³.

6. VCAM1 is expressed in normal Bowman's capsules (PEC) in humans. Figure 6E shows the VCAM1 expression in DTL but not in PEC, and it is also unclear if TNFalpha induced the VCAM1 expression in DTL.

Our additional stainings of human kidney tissue confirmed the reviewer's comment that VCAM1 is expressed in PECs as shown above in Reviewer Fig. 10 (Reviewer 1, item #6) (now included in Suppl. Fig. 8B). We could not confirm VCAM1 expression in PECs in our human kidney single nuclear RNA-seq dataset, as PECs are absent owing to known sample prep limitations. However, our analysis revealed that VCAM1 is differentially expressed in DTL cells in individuals with kidney disease (Fig. 6E). To confirm VCAM1 protein expression in the DTL, we performed additional immunofluorescence studies in kidney tissue from individuals with kidney disease as shown above in Reviewer Fig. 11 (Reviewer 1, item #6), demonstrating VCAM1 expression in DTL particularly in those patients (pts #2, #4) with low GFR (Fig. 6F, Suppl. Fig. 8C). VCAM1 was not observed in DTL in patients without disease (i.e., tumor nephrectomies), (Suppl. Fig. 8A). Relevant text was updated.

7. While the NEPTUNE data is from FSGS/MCD patients, the current manuscript does not show podocyte phenotypes in organoids treated with TNFalpha. Hence, it is unclear if the organoid model recapitulates the patient pathology of FSGS/MCD.

Thank you for raising this important point.

FSGS/MCD is classically described as podocytopathy characterized by proteinuria, alterations of slit diaphragm architecture, and podocyte effacement, as well as scar formation in the glomerulus. While the lack of glomerular-specific vasculature and filtrate flow limits organoid modeling of a mature glomerulus with a mature filtration barrier, organoid modeling affecting podocyte-specific genes has revealed key pathomechanistic insights including for NPHS1-associated disease²⁴⁻²⁷. Our protocol generates transcriptional evidence of podocytes and nephron structures in line

with other protocols¹⁶, and podocytes with a defined, yet immature ultrastructure²⁴. Within this context, our integrated datasets do provide evidence for podocyte injury in kidney organoid podocytes.

First, we see transcriptional evidence of TNF-induced de-differentiation of podocytes, as shown in *Reviewer Fig. 16* (now included in *Suppl. Fig. 4D*). Relevant text was updated.

Reviewer Fig. 16: Dot plot showing decrease in expression of classic podocyte marker genes in TNFalpha-stimulated organoid podocytes.

We also see evidence in our proteomics data of a TNFalpha-induced injury response in podocytes, including altered ITGA3 (increased) and cofilin (decreased) expression (*Suppl. Table 5*). ITGA3 inhibition protects from podocyte injury in vitro²⁸. Other studies find that ITGA3 is linked to podocyte injury²⁹. As shown below in *Reviewer Fig. 17* (and *12 above*), immunostaining reveals localization of ITGA3 within TNF-treated organoid podocytes, among other cells (now included in *Suppl. Fig. 4A*); relevant text was updated. Additionally, a TNF-alpha induced reduction of Cofilin occurs in mice treated with Doxorubicin³⁰. Reduction of this podocyte-enriched protein leads to proteinuria in zebrafish and is also reduced in human/mice with proteinuria³¹.

Reviewer Fig. 17: ITGA3 expression in TNF-treated organoids.

Minor:

1. In lines 144-163, please write the time points of organoids assessed in this section.
 - **This analysis used all detected proteins in organoids from day 21 to 29; we have clarified this in the manuscript.**
2. In lines 165-167, the cited paper (ref # 24) is not a peer-reviewed paper.
 - **This preprint reference which is now published; the References section was updated and the ref. #24 remains the same.**
3. In line 183, figure 4E?
 - **Thank you for noting this discrepancy. We were indeed referring to Fig. 4E, this has been corrected in the text.**
4. In line 199, Fig. 4G?
 - **Thank you for noting this discrepancy. We were indeed referring to Fig. 4G, this has been corrected in the text.**

REVIEWER 3

Lassé et al have conducted a study that combines proteomics and transcriptomics (in bulk and single cell) to study the gene expression changes of kidney organoids during extended culture and disease modeling. Through differential gene analysis, they showed, in both mRNA and protein levels, that extended culture leads to loss of kidney cell signature and accumulation of extracellular matrix in kidney organoids. By comparing the protein profiles between organoids and the adult kidney or immortalized cell line, the authors demonstrate that the kidney organoid is a better cell source than the immortalized cell line for use in kidney disease modeling since it captures more kidney cell type signatures **than the immortalized cell line does**. Then the authors turn to use scRNA-seq and bulk proteomics to assess the disease modeling capacity of kidney organoids with TNF- α treatment. The authors identified a set of TNF- α responsive genes from kidney organoid and demonstrate that this gene-set can be used to as biomarkers to classify the human samples from kidney diseases.

Overall, the proteomics datasets present by the authors in this manuscript can serve as good resources to the scientific community since not many protein profiling datasets are available in the public database. The idea to use proteomics combined with bulk RNA-seq and scRNA-seq to monitor the organoid differentiation and assess disease modeling is quite novel and exceptional. However, I have several concerns about data interpretation and analysis as specified below. Hopefully these comments can help the authors improve their manuscript in the current form.

Thank you for your thoughtful and constructive critique of our manuscript.

1) Kidney organoids hold great promise for disease modeling but its use for that purpose also suffers several drawbacks. For example, the key differences between kidney organoid and adult human kidney are that kidney organoid cell types are not mature and that kidney organoid contains non-kidney cell types such as neurons and muscle cells. When the authors interpret the data from the analysis, they should keep in mind that these differences can perplex the data interpretation especially when the proteomics dataset is from bulk profiling (not single cell) where the gene expression represents an average expression of all cell types including the immature cells and off-target cells.

I suggest the authors to perform additional analyses to help clarify this concern. For example, when the authors compare the proteomics between kidney organoid and adult kidney cells (microdissected tubule and glomeruli), do they see the more developmental genes or neuronal genes expressed in the kidney organoid? Are the off-target cells such as neurons and muscle cells also respond to TNF- α treatment (based on the scRNAseq data).

We have carried out the requested analyses. As we show in *Reviewer Fig. 18* below, there were 188 neuronal proteins (based on uniprot keyword/GO-term annotation for the word 'neuronal') and 299 developmental proteins (based on uniprot keyword/GO-term annotation for the word developmental). 113 neuronal proteins and 244 developmental proteins (19 proteins that fit both neuronal and developmental) were exclusively detected in organoids, together accounting for approximately 7.3% of detected proteins. In both microdissected human kidney tubules and glomeruli, 5.3% and 4.3% respectively were in this category. Not surprisingly, expression of more developmental genes were detected in organoids compared to adult kidney tissues.

Reviewer Fig. 18: Developmental and neuronal proteins detected in organoids compared to human kidney tissue.

Regarding response of off-target organoid cell types to TNFalpha, this indeed is clearly the case as seen for our 10 TNF signature overlap genes in *Suppl. Fig. 5D* (moved position in revised manuscript). This observation prompted our early strategy: 1) to limit the off-target cell type contribution by performing studies on isolated organoid spheroids as opposed to whole wells containing more off-target cell types¹³; and 2) to spatially confirm protein expression by immunofluorescent staining of organoids for target proteins.

2) There are no human kidney diseases solely caused by TNF- α . That raises a question as to whether the use of TNF- α treatment in organoid is a good approach to model kidney diseases. To establish the link between TNF- α and kidney disease, the authors should at least show that TNFalpha signaling (TNF- α and its downstream targets) is upregulated in most forms of the kidney diseases.

Given the complex signaling milieu of the diseased human kidney, it has been extremely challenging teasing out the components that are contributing to kidney disease, especially with rare diseases such as FSGS/MCD. We agree with the reviewer that it is likely that TNFalpha is one proinflammatory component of several contributing to disease activity. Moreover, it is possibly confined to certain stages or to a subset of individuals within the disease spectrum. Indeed, this is what we described in our recent manuscript and previously cited preprint^{11,12} and described by others³². Using a reductionist approach (i.e. treating organoids with TNFalpha), we aim to define a recognizable gene expression pattern that we can associate with stressor activity (such as TNFalpha) which may assist classifying an individual's kidney disease activity.

With this in mind, we refer the reviewer to *Suppl. Fig. 5B* (moved position in revised manuscript) in which we examine TNF activity based on the organoid protein-based TNF signature in several kidney diseases represented in the ERCB dataset. There is a clear trend: tissue from living donors (LD) exhibit the lowest scores, while tissue from inflammation-associated diseases (DKD, RPGN, FSGS/MCD) exhibit higher scores. We see wide error bars for each disease category, suggesting a spectrum of activity even within a disease as observed for combined NEPTUNE MCD/FSGS cohort in *Fig. 5E* (gray bar). When these kidney disease samples were analyzed together, we find increased regulation of the TNF-dependent genes in diseased tissue relative to living donors; we have moved these data from supplemental figure to main *Fig. 5B*. Together, this analysis suggests that TNFalpha activity is involved in pathomechanisms of many forms of kidney disease. Relevant text was updated.

3) The most intriguing point from proteomics data is that the correlation between protein and RNA expression is only less than 0.25, which means many genes are discordant in their RNA and protein levels. The authors should show a heatmap to highlight the differential genes that are only detected in protein but not in mRNA.

We agree, overall correlation is moderate. This is a common value when proteins and transcript data are acquired from the same sample, and has been found in numerous systems, especially very dynamic systems³³. We did not identify any proteins in our proteomics data that are not also detected in our transcriptional data. However, we have

plotted proteins that are only regulated on proteomic, but not transcriptional level; these data are now included in *Suppl. Fig. 2A* and shown below. Relevant text was updated.

a

Reviewer Fig. 19: Genes exhibiting discordant protein and RNA expression in kidney organoids.

4) In human kidney diseases, Vcam1 and C3 are upregulated in the PT and immune cells not in stroma. In Figure 6C, both Vcam1 and C3 are upregulated in stroma cells from kidney organoid. Would that indicate the difference between organoid and adult kidney in terms of cell type responses?

We see in *Fig. 6E* that both C3 and VCAM1 expression are also found in fibroblast and endothelial cells in snRNA-seq of NEPTUNE kidney tissue. We, as well as others, have previously demonstrated that organoid stromal cells express early markers of endothelial cells, mesangial cells, pericytes and fibroblasts^{2,34}. So, it is possible that we are capturing expression in immature versions of these cell types in organoids. We have emphasized this discrepancy now in the revised Discussion, highlighting the limitations of the organoids.

The implications of cell-type specific responses are intriguing but unclear. Detection of a relevant panel capturing gene activity cascade emanating from any kidney cell type may be sufficient to identify a maladaptive process and drive intervention. Clearly the current organoid modeling limitations dictate a very cautious approach when modeling kidney disease, necessitating rigorous benchmarking of any potential organoid model to diseased human biosamples and tissue.

5) Figure 6E shows that DTL responds to TNF- α treatment strongly. Since no much are known for the role of DTL in kidney disease, the authors should do additional work to validate this result. E.g. immunostaining of TNF receptor and the DTL marker to validate that this cell type can respond to TNF- α signals.

We agree that this is important and share the following lines of evidence for expression of TNFR1 in what we identify as the DTL:

1. We performed immunostainings of the TNF α effector VCAM1 in human kidney tissue (see *Reviewer Fig. 11* above under Reviewer 1 item #6) and found that it was increased in the DTL of tissue from individuals with FSGS (now included in *Fig. 6F* and *Suppl. Fig. 8C*). Relevant text was updated. We did not succeed in staining biopsies for TNF receptor.
2. We also revisited the NEPTUNE single nuclear transcriptional dataset and evaluated for expression of markers of DTL, according to the HuBMAP ASCT+B table for Kidney v1.2³⁵. The left dot plot panel in *Reviewer Fig. 20* below shows expression of anticipated marker genes in the annotated DTL cells. The right dot plot panel focuses on DTL cell type expression of TNFRSF1A and effector genes highlighted in this manuscript; notably, the transcript level of each is increased in the high TNF activity tissue, in line with our proteomics analysis of TNF-stimulated organoids.

Reviewer Fig. 20: A) Expression of DTL marker genes in human kidney tissue snRNA-seq cell clusters and B) expression of TNFRSF1A effector genes in this manuscript in DTL cells separated by tissue TNF activity status.

6) In Figure 5e, there are 322 genes in total but when breaking down by secreted and non-secreted proteins, the authors reported 302 + 22 = 324 genes. What are the additional 4 genes from?

We identified 302 proteins in organoid cell lysates & 22 proteins in media that were differentially expressed in TNFalpha-treated kidney organoids; 2 proteins were found in both spaces, for a total of 322 unique proteins. 2 peptides did not initially map to genes, however using updated annotations, we have now expanded the Organoid TNF signature to encompass 322 genes (see updated Fig. 5A, Reviewer Fig. 8-9 above). Relevant text, figures and legends were updated.

7) In Figure 5g, the authors should repeat the analysis by using same number of cells to rule out that the cell response was skewed due to unequal n of cells contributed by each group. I raised this concern because I noticed that the most responsive cell types are the cell types have greater difference in cell number between groups (for example, 2600 vs 30 cells for the immune cell type).

We agree with the reviewer that a differentially abundant cell population between groups could skew comparison of gene expression between cell clusters. This may arise from technical challenges related to tissue procurement and processing of human kidney biopsy samples. However, the response of cells exposed to stimuli may also alter gene expression to such an extent that the clustering algorithm generates separate clusters or sub-clusters complicating direct comparisons, as has been described previously³⁶.

Regarding CXCL10 expression in particular, this does not appear to impact our interpretation: as shown in the track plot below in Reviewer Fig.21 (A), CXCL10 is not expressed at any detectable level in any cell in the group of low TNF snRNA-seq samples; i.e. strikingly, all of the CXCL10 expressing cells are found in the high TNF group. However, as this could impact our interpretation of VCAM1 or C3 expression, we also examined expression in similar numbers of cells in proximity to DTL cells by focusing on PT cells. In (B), expression of VCAM1 and C3 are increased in all high TNF cell types, but more so in DTL (4-fold) > PT + DTL (2-fold) when accounting for differences in mean expression between the dot plots. However, importantly, the relative expression of VCAM1 and C3 in TNF high vs. TNF low cells is very similar when examining DTL in combination with PT cells, adding > 38K cells to the 120 DTL cells.

Reviewer Fig. 21: A) Expression of CXCL10 is detected only in cells from high TNF activity tissue and B) VCAM1, C3 and CXCL10 expression remains low in low TNF activity cells when surveying similar numbers of cells.

8) The authors need to explain the jargons used in the manuscript. For example, what are 2D keyword enrichment and the combined score in the pathway analysis?

Thank you, we have clarified these and other terminologies used in the text, in the methods and figure legends.

“Keyword” is short for uniprot keyword, www.uniprot.org, which is a more concise version of the pathway.

“Combined score” is the output from the package that considers both enrichment and p-value.

References

- Duffield JS. Cellular and molecular mechanisms in kidney fibrosis. *The Journal of clinical investigation*. 2014;124:2299-2306. doi: 10.1172/jci72267
- Davis JL, Kennedy C, Clerkin S, Treacy NJ, Dodd T, Moss C, Murphy A, Brazil DP, Cagney G, Brougham DF, et al. Single-cell multiomics reveals the complexity of TGF β signalling to chromatin in iPSC-derived kidney organoids. *Commun Biol*. 2022;5:1301. doi: 10.1038/s42003-022-04264-1
- Morais MRPT, Tian P, Lawless C, Murtuza-Baker S, Hopkinson L, Woods S, Mironov A, Long DA, Gale DP, Zorn TMT, et al. Kidney organoids recapitulate human basement membrane assembly in health and disease. *eLife*. 2022;11:e73486. doi: 10.7554/eLife.73486
- Lemos DR, McMurdo M, Karaca G, Wilflingseder J, Leaf IA, Gupta N, Miyoshi T, Susa K, Johnson BG, Soliman K, et al. Interleukin-1 β Activates a MYC-Dependent Metabolic Switch in Kidney Stromal Cells Necessary for Progressive Tubulointerstitial Fibrosis. *J Am Soc Nephrol*. 2018;29:1690-1705. doi: 10.1681/asn.2017121283
- Yang X, Delsante M, Daneshpajouhnejad P, Fenaroli P, Mandell KP, Wang X, Takahashi S, Halushka MK, Kopp JB, Levi M, et al. TAZ/TEAD complex regulates TGF- β 1-mediated fibrosis in iPSC-derived renal organoids. *bioRxiv*. 2021:2021.2004.2015.440011. doi: 10.1101/2021.04.15.440011
- Cox J, Mann M. 1D and 2D annotation enrichment: a statistical method integrating quantitative proteomics with complementary high-throughput data. *BMC Bioinformatics*. 2012;13:S12. doi: 10.1186/1471-2105-13-S16-S12
- Browaeys R, Saelens W, Saey Y. NicheNet: modeling intercellular communication by linking ligands to target genes. *Nat Methods*. 2020;17:159-162. doi: 10.1038/s41592-019-0667-5
- Jin S, Guerrero-Juarez CF, Zhang L, Chang I, Ramos R, Kuan C-H, Myung P, Plikus MV, Nie Q. Inference and analysis of cell-cell communication using CellChat. *Nature Communications*. 2021;12:1088. doi: 10.1038/s41467-021-21246-9
- Beg AA, Baltimore D. An essential role for NF-kappaB in preventing TNF-alpha-induced cell death. *Science*. 1996;274:782-784. doi: 10.1126/science.274.5288.782

10. Lawrence T. The nuclear factor NF-kappaB pathway in inflammation. *Cold Spring Harb Perspect Biol.* 2009;1:a001651. doi: 10.1101/cshperspect.a001651
11. Mariani LH, Eddy S, AlAkwa FM, McCown PJ, Harder JL, Martini S, Ademola AD, Boima V, Reich HN, Eichinger F, et al. Multidimensional Data Integration Identifies Tumor Necrosis Factor Activation in Nephrotic Syndrome: A Model for Precision Nephrology. *medRxiv.* 2021:2021.2009.2009.21262925. doi: 10.1101/2021.09.09.21262925
12. Mariani LH, Eddy S, AlAkwa FM, McCown PJ, Harder JL, Nair V, Eichinger F, Martini S, Ademola AD, Boima V, et al. Precision nephrology identified tumor necrosis factor activation variability in minimal change disease and focal segmental glomerulosclerosis. *Kidney Int.* 2023;103:565-579. doi: 10.1016/j.kint.2022.10.023
13. Harder JL, Menon R, Otto EA, Zhou J, Eddy S, Wys NL, O'Connor C, Luo J, Nair V, Cebrian C, et al. Organoid single cell profiling identifies a transcriptional signature of glomerular disease. *JCI Insight.* 2019;4. doi: 10.1172/jci.insight.122697
14. Nishinakamura R. Human kidney organoids: progress and remaining challenges. *Nature Reviews Nephrology.* 2019;15:613-624. doi: 10.1038/s41581-019-0176-x
15. Chambers BE, Weaver NE, Wingert RA. The "3Ds" of Growing Kidney Organoids: Advances in Nephron Development, Disease Modeling, and Drug Screening. *Cells.* 2023;12. doi: 10.3390/cells12040549
16. Wilson SB, Howden SE, Vanslambrouck JM, Dorison A, Alquicira-Hernandez J, Powell JE, Little MH. DevKidCC allows for robust classification and direct comparisons of kidney organoid datasets. *Genome Med.* 2022;14:19. doi: 10.1186/s13073-022-01023-z
17. Takasato M, Er PX, Chiu HS, Maier B, Baillie GJ, Ferguson C, Parton RG, Wolvetang EJ, Roost MS, Chuva de Sousa Lopes SM, et al. Kidney organoids from human iPS cells contain multiple lineages and model human nephrogenesis. *Nature.* 2015;526:564-568. doi: 10.1038/nature15695
18. Kumar SV, Er PX, Lawlor KT, Motazedian A, Scurr M, Ghobrial I, Combes AN, Zappia L, Oshlack A, Stanley EG, et al. Kidney micro-organoids in suspension culture as a scalable source of human pluripotent stem cell-derived kidney cells. *Development.* 2019;146. doi: 10.1242/dev.172361
19. Gururaja TL, Yung S, Ding R, Huang J, Zhou X, McLaughlin J, Daniel-Issakani S, Singh R, Cooper RD, Payan DG, et al. A class of small molecules that inhibit TNFalpha-induced survival and death pathways via prevention of interactions between TNFalphaRI, TRADD, and RIP1. *Chem Biol.* 2007;14:1105-1118. doi: 10.1016/j.chembiol.2007.08.012
20. Espín-Palazón R, Traver D. The NF-kB family: Key players during embryonic development and HSC emergence. *Exp Hematol.* 2016;44:519-527. doi: 10.1016/j.exphem.2016.03.010
21. Collins T, Read MA, Neish AS, Whitley MZ, Thanos D, Maniatis T. Transcriptional regulation of endothelial cell adhesion molecules: NF-kappa B and cytokine-inducible enhancers. *Faseb j.* 1995;9:899-909.
22. Neish AS, Williams AJ, Palmer HJ, Whitley MZ, Collins T. Functional analysis of the human vascular cell adhesion molecule 1 promoter. *J Exp Med.* 1992;176:1583-1593. doi: 10.1084/jem.176.6.1583
23. Lee CW, Lin WN, Lin CC, Luo SF, Wang JS, Pouyssegur J, Yang CM. Transcriptional regulation of VCAM-1 expression by tumor necrosis factor-alpha in human tracheal smooth muscle cells: involvement of MAPKs, NF-kappaB, p300, and histone acetylation. *J Cell Physiol.* 2006;207:174-186. doi: 10.1002/jcp.20549
24. Kim YK, Refaeli I, Brooks CR, Jing P, Gulieva RE, Hughes MR, Cruz NM, Liu Y, Churchill AJ, Wang Y, et al. Gene-Edited Human Kidney Organoids Reveal Mechanisms of Disease in Podocyte Development. *Stem Cells.* 2017;35:2366-2378. doi: 10.1002/stem.2707
25. Hale LJ, Howden SE, Phipson B, Lonsdale A, Er PX, Ghobrial I, Hosawi S, Wilson S, Lawlor KT, Khan S, et al. 3D organoid-derived human glomeruli for personalised podocyte disease modelling and drug screening. *Nat Commun.* 2018;9:5167. doi: 10.1038/s41467-018-07594-z
26. Tanigawa S, Islam M, Sharmin S, Naganuma H, Yoshimura Y, Haque F, Era T, Nakazato H, Nakanishi K, Sakuma T, et al. Organoids from Nephrotic Disease-Derived iPSCs Identify Impaired NEPHRIN

- Localization and Slit Diaphragm Formation in Kidney Podocytes. *Stem Cell Reports*. 2018;11:727-740. doi: <https://doi.org/10.1016/j.stemcr.2018.08.003>
27. Jansen J, van den Berge BT, van den Broek M, Maas RJ, Daviran D, Willemsen B, Roverts R, van der Kruit M, Kuppe C, Reimer KC, et al. Human pluripotent stem cell-derived kidney organoids for personalized congenital and idiopathic nephrotic syndrome modeling. *Development*. 2022;149. doi: 10.1242/dev.200198
 28. Reiser J, Oh J, Shirato I, Asanuma K, Hug A, Mundel TM, Honey K, Ishidoh K, Kominami E, Kreidberg JA, et al. Podocyte migration during nephrotic syndrome requires a coordinated interplay between cathepsin L and alpha3 integrin. *J Biol Chem*. 2004;279:34827-34832. doi: 10.1074/jbc.M401973200
 29. Nicolaou N, Margadant C, Kevelam SH, Lilien MR, Oosterveld MJ, Kreft M, van Eerde AM, Pfundt R, Terhal PA, van der Zwaag B, et al. Gain of glycosylation in integrin $\alpha 3$ causes lung disease and nephrotic syndrome. *The Journal of clinical investigation*. 2012;122:4375-4387. doi: 10.1172/jci64100
 30. Koehler S, Kuczkowski A, Kuehne L, Jungst C, Hoehne M, Grahammer F, Eddy S, Kretzler M, Beck BB, Hohfeld J, et al. Proteome Analysis of Isolated Podocytes Reveals Stress Responses in Glomerular Sclerosis. *J Am Soc Nephrol*. 2020;31:544-559. doi: 10.1681/ASN.2019030312
 31. Ashworth S, Teng B, Kaufeld J, Miller E, Tossidou I, Englert C, Bollig F, Staggs L, Roberts IS, Park JK, et al. Cofilin-1 inactivation leads to proteinuria--studies in zebrafish, mice and humans. *PLoS one*. 2010;5:e12626. doi: 10.1371/journal.pone.0012626
 32. Chung CF, Kitzler T, Kachurina N, Pessina K, Babayeva S, Bitzan M, Kaskel F, Colmegna I, Alachkar N, Goodyer P, et al. Intrinsic tumor necrosis factor- α pathway is activated in a subset of patients with focal segmental glomerulosclerosis. *PLoS one*. 2019;14:e0216426. doi: 10.1371/journal.pone.0216426
 33. Liu Y, Beyer A, Aebersold R. On the Dependency of Cellular Protein Levels on mRNA Abundance. *Cell*. 2016;165:535-550. doi: 10.1016/j.cell.2016.03.014
 34. Czerniecki SM, Cruz NM, Harder JL, Menon R, Annis J, Otto EA, Gulieva RE, Islas LV, Kim YK, Tran LM, et al. High-Throughput Screening Enhances Kidney Organoid Differentiation from Human Pluripotent Stem Cells and Enables Automated Multidimensional Phenotyping. *Cell Stem Cell*. 2018;22:929-940.e924. doi: <https://doi.org/10.1016/j.stem.2018.04.022>
 35. Jain S, Valerius MT, He Y. HuBMAP ASCT+B Tables. Kidney v1.2 <https://doi.org/10.48539/HBM248.CBJV.556>. In; 2022.
 36. Zhao J, Jaffe A, Li H, Lindenbaum O, Sefik E, Jackson R, Cheng X, Flavell RA, Kluger Y. Detection of differentially abundant cell subpopulations in scRNA-seq data. *Proc Natl Acad Sci U S A*. 2021;118. doi: 10.1073/pnas.2100293118

REVIEWERS' COMMENTS

Reviewer #1 (Remarks to the Author):

The authors have addressed all concerns and critiques with additional analyses included in the manuscript or as supplemental. No additional concerns have been raised.

Reviewer #2 (Remarks to the Author):

Authors nicely responded to my comments. Recent publications (e.g. PMID: 36129975) that attempted to address the current challenges (immaturity/lack of vessels) in organoid research could be added to discussion.

Reviewer #3 (Remarks to the Author):

The authors have provided strong data to clear my concerns.

RESPONSE TO REVIEWER COMMENTS

We thank the reviewers and editors for your thoughtful reading of our revised manuscript. We are pleased to have addressed your concerns with our additional supportive data and analyses.

REVIEWER 1

The authors have addressed all concerns and critiques with additional analyses included in the manuscript or as supplemental. No additional concerns have been raised.

REVIEWER 2

Authors nicely responded to my comments. Recent publications (e.g. PMID: 36129975)¹ that attempted to address the current challenges (immaturity/lack of vessels) in organoid research could be added to discussion.

We have added the publication suggested by Reviewer 2 to our introduction section.

“Additionally, organoids have been proposed as a promising screening tool for therapeutics, as well as a model of virus infection and organ cryopreservation, *especially when combined with high throughput methods and organ-on-a-chip microfluidics that allow mechanical forces to be applied.*”

REVIEWER 3

The authors have provided strong data to clear my concerns.

Reference

1. Hiratsuka K, Miyoshi T, Kroll KT, Gupta NR, Valerius MT, Ferrante T, Yamashita M, Lewis JA, Morizane R. Organoid-on-a-chip model of human ARPKD reveals mechanosensing pathomechanisms for drug discovery. *Sci Adv.* 2022;8:eabq0866. doi: 10.1126/sciadv.abq0866